# Locally Hierarchical Auto-Regressive Modeling for Image Generation

**Tackgeun You**[3,5]*
tackgeun.you@postech.ac.kr

**Saehoon Kim**[4]
shkim@kakaobrain.com

**Chiheon Kim**[4]
chiheon.kim@kakaobrain.com

**Doyup Lee**[4]
doyup.lee@kakaobrain.com

**Bohyung Han**[1,2,3]
bhhan@snu.ac.kr

[1]ECE, [2]IPAI, [3]AIIS, Seoul National University, Korea
[4]Kakao Brain, Korea
[5]CSE, POSTECH, Korea

## Abstract

We propose a locally hierarchical auto-regressive model with multiple resolutions of discrete codes. In the first stage of our algorithm, we represent an image with a pyramid of codes using Hierarchically Quantized Variational AutoEncoder (HQ-VAE), which disentangles the information contained in the multi-level codes. For an example of two-level codes, we create two separate pathways to carry high-level coarse structures of input images using top codes while compensating for missing fine details by constructing a residual connection for bottom codes. An appropriate selection of resizing operations for code embedding maps enables top codes to capture maximal information within images and the first stage algorithm achieves better performance on both vector quantization and image generation. The second stage adopts Hierarchically Quantized Transformer (HQ-Transformer) to process a sequence of local pyramids, which consist of a single top code and its corresponding bottom codes. Contrary to other hierarchical models, we sample bottom codes in parallel by exploiting the conditional independence assumption on the bottom codes. This assumption is naturally harvested from our first-stage model, HQ-VAE, where the bottom code learns to describe local details. On class-conditional and text-conditional generation benchmarks, our model shows competitive performance to previous AR models in terms of fidelity of generated images while enjoying lighter computational budgets.

## 1 Introduction

Deep generative models [1, 2, 3] have become the fundamental basis for synthesizing diverse images of high-resolution from various forms of input supervision, including class conditions [2, 4], text prompts [5, 6], and segmentation maps [7] to name a few. Most approaches in this paradigm are roughly categorized into two groups depending on whether they explicitly maximize data likelihood or not. GANs implicitly estimate underlying data distribution to generate novel and crispy samples, but it typically suffers from a lack of diversity [1]. In contrast, it is widely accepted that likelihood-based models are capable of generating diverse examples, compared to GANs, since they aim to cover all modes of a data distribution [8]. This work focuses on auto-regressive models among many likelihood-based approaches motivated by the existing works validating scalability [5, 9].

Auto-Regressive (AR) models factorize a data likelihood into a product of conditionals in the pre-defined generation order. Since AR models already have mode-coverage property, we only need

---

*This work was done during an internship of this author at Kakao Brain.

36th Conference on Neural Information Processing Systems (NeurIPS 2022).

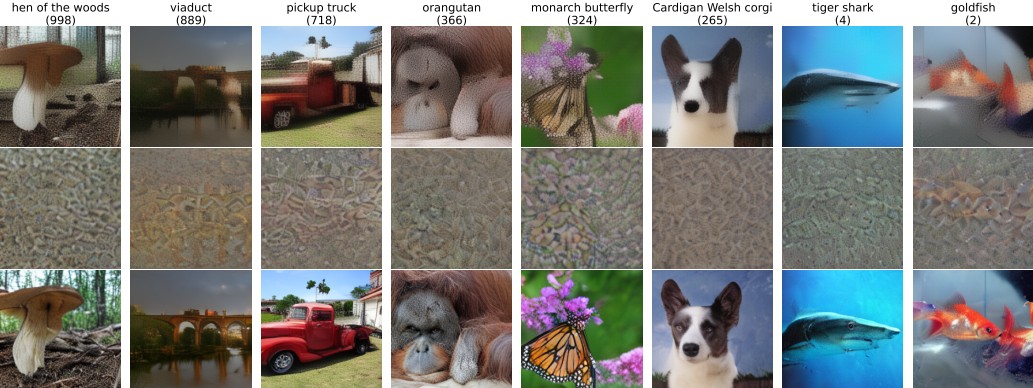

Figure 1: Illustration of generated images (the last row) by our algorithm trained on ImageNet, where the first row shows the examples synthesized by the top codes only while the second row presents the images from the bottom codes only.

to make minimal changes in architectures and optimization methods for the applications to various domains including audio, language, and images. However, regardless of the modality, training AR models in a high-dimensional space is computationally expensive since each conditional needs to consider many past observations. In the context of image generation, VQ-VAE [10] introduces a two-stage learning process, where the first stage represents an image by a discrete code map while the second stage constructs AR priors [2, 11] on discrete sequences. To make this two-stage framework effective, a sequence of the discrete codes has to hold sufficient information for high-quality reconstruction of an original image while the sequence length should be reasonably short for efficient training of AR models.

In this study, we postulate that leveraging the multi-scale structure is a simple but effective way for representing an image as discrete codes with AR modeling. We aim to disentangle the information residing in the hierarchy and design an efficient model to generate unseen images by exploiting the multi-level structure maximally. To achieve this goal, we propose locally hierarchical auto-regressive modeling by extending two-stage AR modeling. We first extend VQ-VAE to produce a two-level pyramid of discrete codes, where the top code contains high-level coarse information and the bottom encodes the low-level fine details. Using these locally hierarchical discrete representations, our efficient AR model generates a novel example. Specifically, the top codes are sequentially generated in a predefined order, and multiple bottom codes associated with a top code are decoded in parallel. The main contributions of the proposed model are summarized below:

- We propose HQ-VAE (Hierarchically Quantized Variational Auto-Encoder), which encodes an image using a pyramid of two-level codes through hierarchical vector quantization. To improve the utility of the two-level codes, we disentangle the information contained in the codes, where top codes capture high-level structural information in images while bottom codes carry the remaining low-level details.

- We introduce HQ-Transformer (Hierarchically Quantized Transformer), which is an AR model to generate images based on hierarchical discrete codes. We predict a sequence of locally hierarchical codes efficiently by maximizing its likelihood based on the conditional independence assumption imposed on bottom codes.

- The proposed algorithm achieves competitive performance on class- and text-conditional image generation tasks while yielding high throughput in sample generation, compared to various types of generative models.

Figure 1 illustrates several examples of generated images by the proposed algorithm. We also present the synthesized images based only on top and bottom codes together, which clearly exhibits the proper disentanglement of image contents and low-level details.

## 2   Related Works

This section describes existing approaches closely related to our algorithm and highlights the differences between the prior works and ours.

**Discrete representations for autoregressive models**   Learning AR priors based on discrete representations turn out to be effective in language modeling [9, 12]. VQ-VAE [10] pioneers the two-stage image generation process, where the first stage represents an image by a set of discrete codes and the next stage constructs AR priors [2, 11] on the discrete sequences. Following VQ-VAE, many approaches aim to further reduce the sequence length and improve the architecture [13, 14] or training objectives [7] of the first stage. Specifically, VQ-GAN shortens the sequence length by using the perceptual and adversarial losses and encoding the semantic information in the codes effectively. However, these approaches still exhibit the fundamental trade-off between code sequence length and generated image quality due to a lack of expressiveness of the single-scale vector quantization with the reduced length of the code sequence.

**Hierarchical representations for autoregressive models**   The construction of hierarchical codes is an effective way to control the fundamental trade-off related to the length of the code sequence. There exist several image representation and generation methods based on hierarchical discrete codes [8, 15, 16]. VQ-VAE-2 [8] and HAM [15] make use of the hierarchical relationship between the codes for unseen image generations in a coarse-to-fine manner. In particular, to tackle the trade-off issue, VQ-VAE-2 [8] allocates a heavier model for processing coarse top codes and a lighter model for dealing with fine bottom codes. RQ-Transformer [17] encodes an image into a stack of code maps by iteratively quantizing residual vectors between latent features and code embeddings, effectively reducing the spatial resolution of the code maps without degrading reconstruction quality. Hierarchical VQ-VAE [16] encodes structural and textual information of an image using different types of codes by training two separate VQ-VAEs, where one learns with the original images and the other employs the residuals between the original and reconstructed images. The main idea of Hierarchical VQ-VAE is about how to disentangle the information across the code levels but not about how to train AR models based on this set of codes. Our approach constructs a pyramid of discrete codes and exploits the codes using AR models to generate high-fidelity images.

**Diffusion models**   Diffusion models [3, 18, 19] iteratively denoise a sample from white noise by seeing the contexts in a bidirectional manner. Most diffusion models have been implemented based on the U-Net [20] style architecture, meaning that the multi-scale feature of an image has been naturally incorporated. However, they suffer from low throughput in the generation process because, even though the inference requires a lot of iterative steps, the intermediate computation results between the denoising steps are not reusable, unlike AR models.

## 3   Locally Hierarchical Auto-Regressive Models

This section describes the details of the proposed method for image generation. Our approach exploits two levels of codes—top and bottom—for the representation of an image contrary to existing two-stage approaches based on AR models. The proposed architecture encourages the top code to model the overall structure of an image while facilitating the bottom code to complement the remaining details of the image. To this end, we propose Hierarchically Quantized Variational AutoEncoder (HQ-VAE) for the image vector quantization in the first stage followed by HQ-Transformer to learn the distribution over a sequence of the top and bottom codes in the second stage. HQ-Transformer achieves better perceptual quality and sampling speed using the underlying hierarchical structure. Figure 2 illustrates the two-stage procedure of our algorithm based on an AR model.

### 3.1   Stage 1: Hierarchically Quantized VAE (HQ-VAE)

We present the architecture and the optimization method of the proposed HQ-VAE, for which we first review the vector quantization technique and then discuss our hierarchical quantization module.

#### 3.1.1   Notations for vector quantization

Vector quantization [21] maps a $d$-dimensional continuous feature vector to a discrete code, which is given by identifying the nearest code embedding of the feature vector in a codebook. The codebook

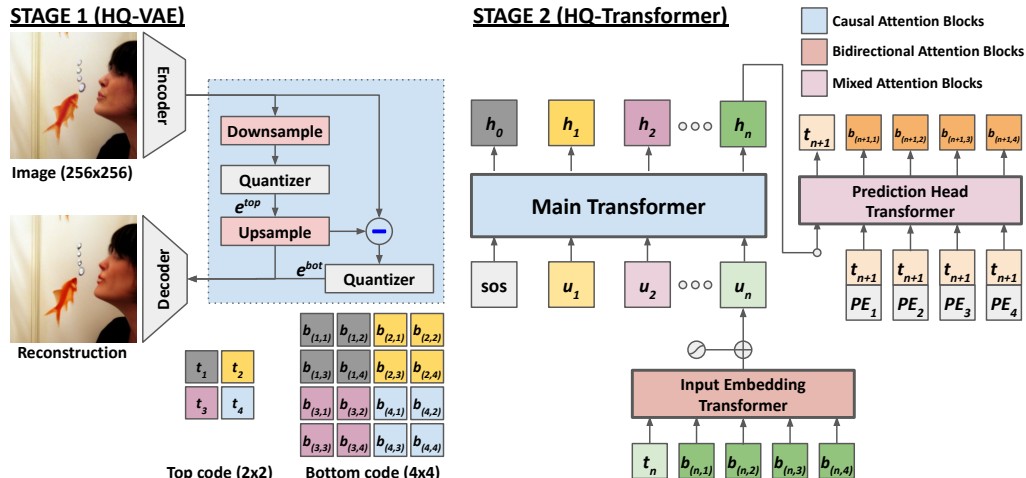

Figure 2: Conceptual illustration of the overall architecture: (a) components of HQ-VAE with residual and resampling operations to encode different information levels in the top and bottom codes, and (b) components of HQ-Transforms to train an AR prior on the top and bottom codes.

is a collection of tuples $\mathcal{C} = \{(s, \mathbf{e}^s)\}$, where $\mathbf{e}_s \in \mathcal{R}^{d_e}$ denotes the embedding of code $s$ and $n_{\mathcal{C}}$ is the size of the codebook $\mathcal{C}$. Supposing that $\mathbf{z}_{ij}$ is a $d$-dimensional vector at a spatial location $(i, j)$ in a feature map $\mathbf{z} \in \mathbb{R}^{l \times l \times d}$, a code $s_{ij}$ is given by

$$s_{ij} \equiv Q(\mathbf{z}_{ij}; \mathcal{C}) = \underset{s}{\operatorname{argmin}} ||\mathbf{z}_{ij} - \mathcal{C}[s]||, \tag{1}$$

where $\mathcal{C}[\cdot]$ returns the code embedding of an input code and $Q(\cdot; \mathcal{C})$ denotes the quantization process based on a codebook $\mathcal{C}$. Note that, since every element in a feature map has its own code, the spatial resolution of the code map, denoted by $\mathbf{s}$, is the same as that of the feature map.

### 3.1.2 Hierarchical vector quantization

Our approach adopts a hierarchical vector quantization scheme to encode input data using two levels of discrete codes—top and bottom denoted by $\mathbf{t}$ and $\mathbf{b}$, respectively. Ideally, the combination of the two-level codes is expected to maintain the full representation power of an original feature map. In addition, we desire the two kinds of codes to have sufficiently disentangled and heterogeneous information; for example, top codes hold the high-level concept of inputs while bottom ones represent remaining low-level details. To these ends, our network creates two separate pathways designated for the top and bottom codes and makes the branch for bottom codes constrained to have residual information. In specific, our hierarchical vector quantization transforms a feature map $\mathbf{z} \in \mathbb{R}^{rl \times rl \times d}$ into a tuple of two code maps $(\mathbf{t}, \mathbf{b})$, where $\mathbf{t} \in \mathcal{Z}^{l \times l}$ and $\mathbf{b} \in \mathcal{Z}^{rl \times rl}$ with an integer scaling factor $r \in \{1, 2, \dots\}$.

Our scheme first captures the high-level information of a feature map by quantizing its downsampled version using the top codes and computing their embeddings as follows:

$$\mathbf{z}^{\text{top}} = \text{Downsample}(\mathbf{z}; r), \qquad t_{ij} = Q^{\text{top}}(\mathbf{z}^{\text{top}}_{ij}; \mathcal{C}^{\text{top}}_{ij}), \qquad \mathbf{e}^{\text{top}}_{ij} = \mathcal{C}^{\text{top}}[t_{ij}], \tag{2}$$

where $\mathcal{C}^{\text{top}}$ denotes the codebook of top codes and $\text{Downsample}(\cdot)$ is a generic pooling operation to reduce the spatial resolution of a feature map. Given the top code map $\mathbf{t}$, the bottom codes and its embeddings are derived by

$$\mathbf{z}^{\text{bot}} = \mathbf{z} - \text{Upsample}(\mathbf{e}^{\text{top}}; r), \quad b_{ij} = Q^{\text{bot}}(\mathbf{z}^{\text{bot}}_{ij}; \mathcal{C}^{\text{bot}}_{ij}), \quad \mathbf{e}^{\text{bot}}_{ij} = \mathcal{C}^{\text{bot}}[b_{ij}], \tag{3}$$

where $\mathcal{C}^{\text{bot}}$ is the codebook of bottom codes and $\text{Upsample}(\cdot)$ is any upsampling operation to increase the spatial resolution of a feature map. Figure 2(left) illustrates this hierarchical quantization procedure. Note that the residual vector between the original feature map and the upsampled top code embedding map, denoted by $\mathbf{z}^{\text{bot}}$, involves sampling and quantization errors. The bottom codes with their embeddings should compensate for the errors and improve the reconstruction quality of the original feature map when combined with the top-level information.

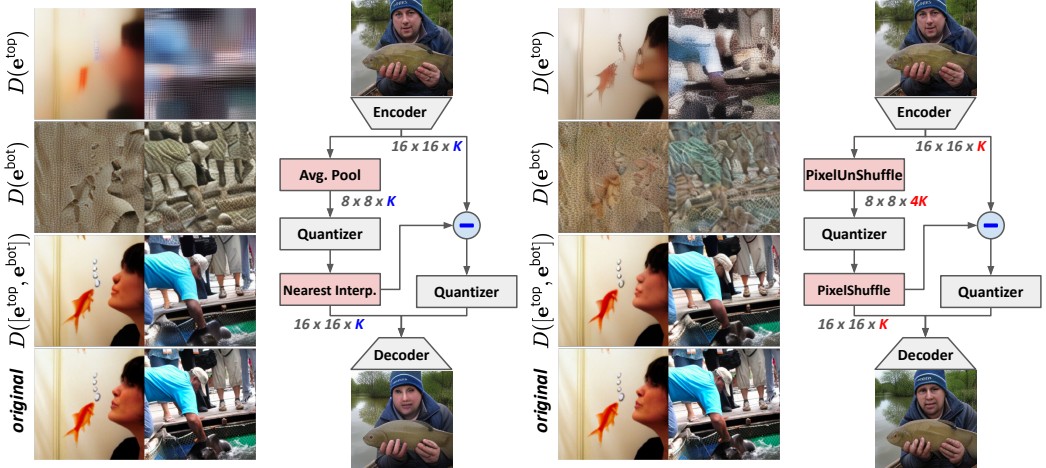

(a) Average pooling and nearest interpolation       (b) Pixel-unshuffle and pixel-shuffle

Figure 3: Comparison of different implementations of resampling operations in our hierarchical vector quantization procedure. When the average pooling and nearest interpolation are used for resizing as in (a), the reconstructed images from the top codes only are too blurry. On the other hand, resizing by pixel-unshuffle and pixel-shuffle operations leads to significantly better reconstruction quality in the same condition as shown in (b).

One question is which operations are appropriate for Downsample($\cdot$) and Upsample($\cdot$). We observe that non-learnable layers give favorable solutions for the resizing operations compared to naïve learnable counterparts. Straightforward choices for those operations are the average pooling and the interpolation with nearest neighbors. Despite the reasonable performance given by this strategy, Figure 3a exhibits that the overall structure reconstructed by the top code embeddings is below expectation. Instead, it turns out that a combination of pixel-unshuffle and pixel-shuffle operations [22] effectively pushes more information into the top codes while making the bottom codes focus on local details. Given the feature map $\mathbf{z} \in \mathbb{R}^{rl \times rl \times d}$ and a scaling factor $r$, the pixel-unshuffle operation returns a tensor in the shape of $l \times l \times r^2 d$, encoding high-level information of the original feature map. To compute the residual, we pixel-shuffle the top code embedding and restore the original dimensionality of $\mathbf{z}$. Figure 3 clearly demonstrates that the top codes given by the pixel-unshuffle followed by the quantization hold the primary information of an input image more than the ones given by the average pooling.

### 3.1.3 HQ-VAE: architecture and optimization

**Overall structure** HQ-VAE consists of three parts: encoder $E(\cdot)$, decoder $D(\cdot)$, and hierarchical quantizer $\mathcal{H}(\cdot; r)$, as shown in Figure 2(left), where $r$ is the additional scaling factor for the top code. Given an image $\mathbf{x}$, an encoder $E(\cdot)$ extracts the feature map $\mathbf{z}$. Then, the hierarchical quantizer $\mathcal{H}(\cdot; r)$ is applied to the feature map, providing the top and bottom codes $(\mathbf{t}, \mathbf{b})$ and their associated code embeddings: $\mathbf{e}^{\text{top}}$ and $\mathbf{e}^{\text{bot}}$. Now, the decoder $D(\cdot)$ takes the top and bottom code embeddings to reconstruct the input image.

**Objective function** To train three parts in HQ-VAE, our objective function is composed of the reconstruction loss introduced in VQ-VAE [10] as well as adversarial and perceptual losses to improve the fidelity of reconstructed images [7]. Here, we only present the reconstruction loss as below:

$$\mathcal{L}_{\text{rec}}(\mathbf{x}; E, D, \mathcal{H}) = \|\mathbf{x} - D([\mathbf{e}^{\text{top}}, \mathbf{e}^{\text{bot}}])\|_2^2 \quad + \sum_{i \in \{\text{top}, \text{bot}\}} \left[ \|\text{sg}(\mathbf{z}^i) - \mathbf{e}^i\|_2^2 + \beta \|\text{sg}(\mathbf{e}^i) - \mathbf{z}^i\|_2^2 \right], \tag{4}$$

where $\text{sg}(\cdot)$ refers to a stop-gradient operation, and $\beta$ is set to 0.25 as suggested in [10]. The first term measures $\ell_2$ distance between the reconstructed and original images. The second term enforces both the top and bottom code embedding to be located close to the inputs of the quantizers. To update the codebook, we make use of exponential moving average (EMA) to accelerate convergence as discussed in [10].

**Implementation details**  We adopt the same network structures of the encoder and the decoder as VQ-GAN [7] except two minor modifications. To increase throughput in training, we set the stride of the first convolution layer in the encoder and the last transposed convolution layer in the decoder as 2, instead of 1 in the original implementation. We replace the batch normalization [23] in the discriminator by the group normalization [24] to achieve consistent results regardless of batch sizes.

## 3.2 Stage 2: Hierarchically Quantized Transformer (HQ-Transformer)

This section presents HQ-Transformer (see Figure 2(right)) to autoregressively model the top and bottom codes by fully exploiting their spatial relationships obtained from the first stage.

### 3.2.1 Formulation

Our objective is to maximize the log-likelihood of a tuple of $\mathbf{t}$ and $\mathbf{b}$. Given the scaling factor $r$ for the top, the resizing operation aligns a single top code $t_i$ with $r^2$ bottom codes $b_{(i,k)}$, where $1 \leq k \leq r^2$. Figure 2 illustrates an example of top and bottom codes with $r = 2$ (in the input embedding transformer part), where all bottom codes aligned with a single top code $t_i$ are $\mathbf{b}_i = (b_{(i,1)}, b_{(i,r^2)})$. The likelihood of the top and bottom codes is given by

$$\mathbb{P}_\theta\left(\mathbf{t}, \mathbf{b}\right) = \mathbb{P}_\theta\left(t_1 \cdots t_{l^2}, b_{(1,1)} \cdots b_{(l^2,r^2)}\right) = \prod_{i=1}^{l^2} \mathbb{P}_\theta\left(t_i, \mathbf{b}_i | \mathbf{t}_{<i}, \mathbf{b}_{<i}\right), \tag{5}$$

where the inequality symbol $(< i)$ denotes a collection of all indices less than $i$. The conditional is further factorized into

$$\mathbb{P}_\theta\left(t_i, \mathbf{b}_i | \mathbf{t}_{<i}, \mathbf{b}_{<i}\right) = \mathbb{P}_\theta\left(\mathbf{b}_i | t_i, \mathbf{t}_{<i}, \mathbf{b}_{<i}\right) \mathbb{P}_\theta\left(t_i | \mathbf{t}_{<i}, \mathbf{b}_{<i}\right). \tag{6}$$

The last term in (6) is maximized by predicting the current position given the previous top and bottom codes. By assuming that the bottom codes are conditionally independent given the associated top code, the first term of the right-hand side is formulated more efficiently as follows:

$$\mathbb{P}_\theta\left(\mathbf{b}_i | t_i, \mathbf{t}_{<i}, \mathbf{b}_{<i}\right) \triangleq \prod_{k=1}^{r^2} \mathbb{P}_\theta\left(b_{(i,k)} | t_i, \mathbf{t}_{<i}, \mathbf{b}_{<i}\right), \tag{7}$$

Note that the assumption we made here is reasonable because the bottom codes are supposed to contain local details while the top codes are enforced by our stage 1 model, HQ-VAE, to encode a global landscape.

### 3.2.2 Model

Similar to GPT-like architectures [9, 25], HQ-Transformer consists of three sub-modules based on the transformer architecture, which are referred to as Input Embedding Transformer (IET), Main Transformer (MT), and Prediction Head Transformer (PHT), as illustrated in Figure 2(right). IET generates an embedding to be used as an input for MT, and the hidden state given by MT is fed into PHT to generate top and bottom codes, which are given back to IET in the next step. We discuss each of the three modules in the rest of this section.

**Input Embedding Transformer (IET)**  IET transforms a tuple of $(t_i, \mathbf{b}_i)$ into a single embedding vector $\mathbf{u}_i$, $\forall i$. To implement this, we first construct a local embedding by summing up the code embedding and the positional encoding, which is given by

$$\begin{aligned}
\mathbf{m}_i^{\text{top}} &= \mathbf{e}_i^{\text{top,in}} + \text{PE}^{\text{top}}[i] + \text{PE}^{\text{bot}}[0] \\
\mathbf{m}_{(i,k)}^{\text{bot}} &= \mathbf{e}_{(i,k)}^{\text{bot,in}} + \text{PE}^{\text{bot}}[k], \qquad \text{for } k = 1, \cdots, r^2,
\end{aligned} \tag{8}$$

where $\mathbf{e}_i^{\text{top,in}}$ and $\mathbf{e}_{(i,k)}^{\text{bot,in}}$ are the learnable code embedding of $t_i$ and $b_{(i,k)}$, respectively, and $\text{PE}^{\text{top}}[\cdot]$ and $\text{PE}^{\text{bot}}[\cdot]$ denote the learnable positional encoding for the top and bottom[2]. Note that the code

---

[2]$\text{PE}^{\text{bot}}[0]$ is introduced to distinguish the top code from the bottom ones.

embeddings are learned within this IET module, by which we construct the codebooks $\mathcal{C}^{\text{top,in}}$ and $\mathcal{C}^{\text{bot,in}}$. These local embeddings are refined by a bidirectional transformer,

$$\mathbf{u}_i^{\text{top}}, \mathbf{u}_{(i,1)}^{\text{bot}}, \cdots, \mathbf{u}_{(i,r^2)}^{\text{bot}} = \text{IET}\left(\mathbf{m}_i^{\text{top}}, \mathbf{m}_{(i,1)}^{\text{bot}}, \cdots, \mathbf{m}_{(i,r^2)}^{\text{bot}}\right), \tag{9}$$

where the structure is the same as GPT-2 [9] and we employ a single GPT-2 block composed of a self-attention followed by an MLP block. Then, the final input embedding $\mathbf{u}_i$ for the main Transformer block is simply given by averaging all updated local embeddings as

$$\mathbf{u}_i = \text{Average}(\mathbf{u}_i^{\text{top}}, \mathbf{u}_{(i,1)}^{\text{bot}}, \cdots, \mathbf{u}_{(i,r^2)}^{\text{bot}}). \tag{10}$$

**Main Transformer (MT)** This module updates the outputs of IET by a stack of $N_{\text{MT}}$ causal self-attention and MLP blocks:

$$\mathbf{h}_0, \mathbf{h}_1, \cdots, \mathbf{h}_{l^2-1} = \text{MT}(\text{SOS}, \mathbf{u}_1, \mathbf{u}_2, \cdots, \mathbf{u}_{l^2-1}), \tag{11}$$

where the sequence length is equal to the number of top codes and the start-of-sentence token (SOS) is employed as the first input of the $\text{MT}(\cdot)$ function. This function also has the same structure as GPT-2.

**Prediction Head Transformer (PHT)** This module predicts the top and bottom codes from the hidden representation derived by the main transformer, where the bottom codes are generated in parallel based on their associated top code and a hidden representation from MT. In specific, we first define the input sequence of PHT as below:

$$\mathbf{v}_i^{(k)} = \begin{cases} \mathbf{h}_i + \text{PE}^{\text{head}}(k) & \text{if} \quad k = 0, \\ \mathbf{e}_i^{\text{top,head}} + \text{PE}^{\text{head}}(k) & \text{if} \quad 1 \leq k \leq r^2, \end{cases} \tag{12}$$

where $\mathbf{e}_i^{\text{top,head}}$ denotes the learnable code embedding of $t_i$ defined within PHT and $\text{PE}^{\text{head}}[\cdot]$ denotes the learnable positional encoding for top and associated bottom codes defined within PHT. This sequence is then updated to produce the conditional probabilities of the top and bottom codes as follows:

$$p_{i+1}^{(0)}, p_{i+1}^{(1)}, \cdots, p_{i+1}^{(r^2)} = \text{PHT}(\mathbf{v}_i^{(0)}, \mathbf{v}_i^{(1)}, \cdots, \mathbf{v}_i^{(r^2)}), \tag{13}$$

where each output probability is defined as

$$p_i^{(k)} = \begin{cases} \mathbb{P}\left(t_i | \mathbf{t}_{<i}, \mathbf{b}_{<i}\right) & \text{if} \quad k = 0, \\ \mathbb{P}\left(b_{(i,k)} | t_i, \mathbf{t}_{<i}, \mathbf{b}_{<i}\right) & \text{if} \quad 1 \leq k \leq r^2 \end{cases}. \tag{14}$$

Note that we predict the bottom code distributions in parallel. In PHT, we employ a self-attention block of a mixture of the positions of a top code and multiple bottom codes, which is conditioned on the top position while the attention between bottom positions is unconditional. Compared to the full autoregressive modeling on the bottoms, such a parallel decoding scheme for bottom codes leads to a speed-up in sampling and even better image generation quality. We will discuss its benefits in the experiments section.

### 3.2.3 Computational complexity of HQ-Transformer

Given the top code $\mathbf{t} \in \mathcal{Z}^{l \times l}$ and the bottom code $\mathbf{b} \in \mathcal{Z}^{rl \times rl}$, where $r$ is a scaling factor for the quantization of top codes, the length of the top codes is $L_{\text{top}} = l^2$ and the number of codes in a local patch is $L_{\text{local}} = 1 + r^2$ because there exist $r^2$ bottom codes associated with a single top code. The time complexities of IET, MT, and PHT are then given by $O(N_{\text{IET}}L_{\text{top}}L_{\text{local}}^2)$, $O(N_{\text{MT}}L_{\text{top}}^2)$ and $O(N_{\text{PHT}}L_{\text{top}}L_{\text{local}}^2)$, respectively, where $N_{\text{IET}}$, $N_{\text{MT}}$, and $N_{\text{PHT}}$ are the numbers of attention blocks in individual modules. The time complexity of MT dominates those of the other two modules because $N_{\text{MT}}$ is larger than $N_{\text{IET}}$ and $N_{\text{PHT}}$. Because the code map size of VQ-GAN is equal to the bottom code resolution of HQ-Transformer, the computational complexity of VQ-GAN, $O(N_{\text{MT}}L_{\text{top}}^2 r^4)$, is higher than the complexity of HQ-Transformer, $O(N_{\text{MT}}L_{\text{top}}^2)$.

Table 1: Class-conditional image generation performance on ImageNet in terms of FID, precision, recall, and throughput. The values in parentheses are obtained by rejection sampling.

| Models | Parameter size | Code map resolution | FID $\downarrow$ | Precision $\uparrow$ | Recall $\uparrow$ | Throughput $\uparrow$ # images/sec |
|---|---|---|---|---|---|---|
| ADM [26] | 554 M | - | 10.94 (4.59) | 0.69 (0.82) | 0.63 (0.52) | 0.12 (0.07) |
| ImageBART [27] | 3.5 B | $16 \times 16$ | 21.19 (7.44) | - | - | - |
| VQ-Diffusion [19] | 370 M | $32 \times 32$ | 11.89 (5.32) | - | - | 2.66 (0.13) |
| LDM-4 [18] | 400 M | $64 \times 64$ | 10.56 (3.60) | 0.71 (0.87) | 0.62 (0.48) | 0.70 (0.30) |
| MaskGIT [28] | 228 M | $16 \times 16$ | 6.18 (4.02) | 0.80 | 0.51 | 11.04 (0.55) |
| VQ-VAE-2 [8] | 13.5 B | $32 \times 32 + 64 \times 64$ | $\sim 31$ ($\sim 10$) | 0.36 | 0.57 | - |
| ViT-VQGAN [13] | 1.6 B | $32 \times 32$ | 4.17 (3.04) | - | - | - |
| VQ-GAN [7] | 1.4 B | $16 \times 16$ | 15.78 (5.20) | - | - | 6.76 (0.34) |
| RQ-Transformer [17] | 1.4 B | $8 \times 8 \times 4$ | 8.71 (3.89) | 0.71 | 0.58 | 37.03 (1.85) |
| RQ-Transformer [17] | 3.8 B | $8 \times 8 \times 4$ | 7.55 (3.80) | 0.73 (0.82) | 0.58 (0.50) | 23.22 (2.90) |
| HQ-TVAE (S) | 530 M | $8 \times 8 + 16 \times 16$ | 9.36 (4.98) | 0.69 (0.78) | 0.55 (0.51) | 78.87 (19.72) |
| HQ-TVAE (M) | 870 M | $8 \times 8 + 16 \times 16$ | 8.46 (4.47) | 0.71 (0.79) | 0.56 (0.52) | 70.00 (17.50) |
| HQ-TVAE (L) | 1.4 B | $8 \times 8 + 16 \times 16$ | 7.15 (4.35) | 0.73 (0.81) | 0.55 (0.50) | 49.44 (12.36) |

Table 2: Text-conditional image generation performance on the CC3M validation set.

| Models | Parameter size | Code map resolution | FID $\downarrow$ | CLIP-score $\uparrow$ | Throughput $\uparrow$ |
|---|---|---|---|---|---|
| ImageBART [27] | 2.8 B | $16 \times 16$ | 22.61 | 0.23 | 1.75 |
| LDM [18] | 645 M | $64 \times 64$ | 17.01 | 0.24 | 3.9 |
| VQ-GAN [7] | 1.5 B | $16 \times 16$ | 28.86 | 0.20 | 2.69 |
| RQ-Transformer [17] | 654 M | $8 \times 8 \times 4$ | 12.33 | 0.26 | 46.47 |
| HQ-TVAE (S) | 579 M | $8 \times 8 + 16 \times 16$ | 12.86 | 0.26 | 54.88 |

### 3.3 Sampling from HQ-Transformer and HQ-VAE

For inference, we first generate discrete codes using HQ-Transformer and synthesize new images using HQ-VAE based on the codes. We begin with the start-of-sentence (SOS) token and obtain its hidden representation from MT in HQ-Transformer, where the class-conditional SOS is employed in the case of class-conditional generation. Then, PHT takes the hidden representation as its input embedding at the first step and predicts a top code distribution. We sample a top code from the distribution and forward it to the next step of PHT to obtain the bottom codes, which are actually acquired in parallel by forwarding the top code embedding to the multiple next steps. The top and bottom codes are given to IET, which outputs the embedding for the input to MT. We repeat this procedure in an auto-regressive manner until we obtain all the top and bottom codes. Once completing the generation of the codes, we finally produce a synthetic image using the decoder of HQ-VAE based on the discrete codes.

## 4 Experiments

We evaluate our models, referred to as HQ-TVAE hereafter, on class- and text-conditional image generation tasks.

### 4.1 Conditional Image Generation

**Experiment settings** We train our models on 1.2M images in the train split of ImageNet [29] for class-conditional image generation. For text-conditional tasks, we employ 15M image-text pairs in Conceptual Caption (CC) [30] and Conceptual-12M [31]. In the training phase of both cases, we resize the short axis of an image to 256 and randomly crop it to acquire a $256 \times 256$ image. In testing, we also resize the short axis to 256 and crop the image at the center.

The scaling factor in HQ-VAE is set to 2, *i.e.*, $r = 2$, to produce the top and bottom codes of $8 \times 8$ and $16 \times 16$ resolution by default, respectively. We test three versions of HQ-Transformer by varying hyperparameters: (a) $N_{\mathrm{MT}} = 12$, and $N_{\mathrm{PHT}} = 4$ for the smallest model, (b) $N_{\mathrm{MT}} = 24$ and $N_{\mathrm{PHT}} = 4$ for the mid-size model, and (c) $N_{\mathrm{MT}} = 42$, and $N_{\mathrm{PHT}} = 6$ for our largest model. In all cases, $N_{\mathrm{IET}} = 1$ and other parameters related to the network size are fixed. We describe more details about the network architecture in the supplementary document. As evaluation metrics, we adopt Fréchet Inception Distance (FID) [32] and precision & recall (PR) [33] for class-conditional generation, and

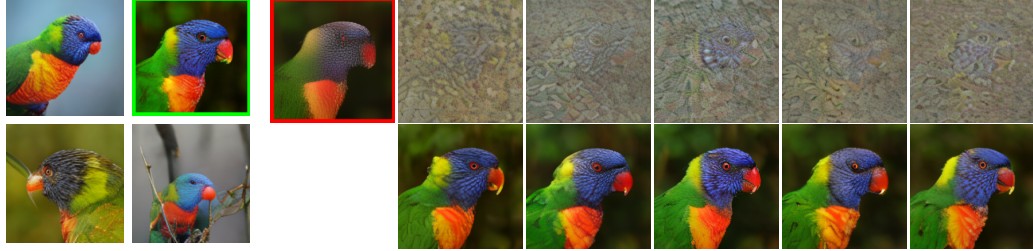

|  (a) Generated samples  |  (b) Top code-conditional image generation  |
|:---:|:---:|

Figure 4: Class-conditional image generation results. Images constructed by both top and bottom codes for an ImageNet class *lorikeet* (90) are illustrated in (a). Given an image generated from the top code only, the left-most image in (b) (with a red bounding box), which corresponds to the upper-right image in (a) (with a green bounding box), HQ-Transformer samples several different bottom codes visualized at the first row in (b) and generates realistic images with diverse details as shown in the second row in (b).

Table 3: Comparison of image reconstruction quality on the ImageNet validation set. RQ-VAE* is a modified version of its original model obtained by adopting the same architecture used in HQ-VAE.

| Model | Code map resolution | rFID ↓ | Codebook size | Effective code usage |
|---|---|---|---|---|
| DALL-E [5] | $32 \times 32$ | 32.01 | 8192 | - |
| AugVAE-SL [14] | $32 \times 32$ | 3.28 | 8192 | - |
| ViT-VQGAN [13] | $32 \times 32$ | 1.28 | 8192 | - |
| VQ-GAN [7] | $16 \times 16$ | 4.90 | 16384 | 973 |
| RQ-VAE [17] | $8 \times 8 \times 4$ | 3.20 | 16384 | 16353 |
| RQ-VAE* [17] | $8 \times 8 \times 5$ | 3.37 | 16384 | 16322 |
| HQ-VAE ($16 \times 16$) | $8 \times 8 + 16 \times 16$ | 2.61 | $8192 + 8192$ | $8119 + 8192$ |
| HQ-VAE ($32 \times 32$) | $16 \times 16 + 32 \times 32$ | 0.57 | $8192 + 8192$ | $8192 + 8192$ |

FID and CLIP-score [34] for the text-conditional generation task. We measure the throughput of sample generation on a single Tesla A100 GPU.

**Class-conditional image generation**    Table 1 presents that our largest model, HQ-TVAE (L), achieves the best FID (7.15) compared to other two-stage AR models with similar sequence lengths such as VQ-GAN ($16 \times 16$) and RQ-Transformer ($8 \times 8 \times 4$). We observe that HQ-TVAE reports the highest throughput in sample generation than other models because it generates $r^2$ bottom codes in parallel given an associated top code. Compared to ADM [26], our smallest model (S) generates images of similar quality while achieving 650 times throughput. We also conduct rejection sampling in all the models based on ResNet-101 [35] with an acceptance rate of 0.25 and report the results within the parentheses in Table 1. Figure 4 demonstrates the qualitative results of our approach, which also include the generated images based only on top codes to visualize the representation disentanglement. Interestingly. contrary to typical generative models, HQ-TVAE restricts the diversity of generated images within details by sampling only bottom codes from HQ-Transformer with fixed top codes.

**Text-conditional image generation**    Table 2 presents FIDs and CLIP-scores of several models measured on the CC3M validation split, and the throughput in sample generation. The results show that our model is comparable to RQ-Transformer in terms of FID (12.86 vs. 12.33) and CLIP-score (0.26 vs. 0.26) while achieving 18%p higher throughput than RQ-Transformer.

## 4.2    Analysis of HQ-VAE and HQ-Transformer Models

**Comparison with other stage 1 methods**    Table 3 compares HQ-VAE implementations with two different feature map sizes (shown in the parentheses) from the encoder with the existing stage 1 models, where we measure FIDs between the original and reconstructed images, referred to as rFID. The last column of the table reports the utilization of codebook(s) for the representations of the images in the ImageNet validation dataset, and our algorithm turns out to make full use of the codebooks. rFID from HQ-VAE ($8 \times 8 + 16 \times 16$) is better than the baselines with the similar sequence length, *e.g.*, RQ-VAE ($8 \times 8 \times 4$) and RQ-VAE ($8 \times 8 \times 5$). In addition, HQ-VAE ($16 \times 16 + 32 \times 32$)

Table 4: Ablation study on HQ-VAE.

| Downsampling op. | Epochs | rFID ↓ |
|---|---|---|
| Conv2 + Perceptual loss | 15 | 3.72 |
| Avg. Pool. | 15 | 3.67 |
| PixelUnshuffle | 15 | 3.14 |
| PixelUnshuffle | 50 | 2.61 |

Table 5: Ablation study on HQ-Transformer

| Decoding policy | Label type | FID ↓ |
|---|---|---|
| Global conditioning | One-hot | 23.25 |
| Locally sequential conditioning | One-hot | 11.33 |
| Locally hierarchical conditioning | | 10.01 |
| Locally hierarchical conditioning | Soft ($\tau = 1$) | 9.36 |

Table 6: Performance of HQ-TVAE with more than two levels of codes. The performance is evaluated on the ImageNet validation split. HQ-VAEs and HQ-Transformers are trained for 15 and 100 epochs respectively for a fair comparison.

| Code map resolution | rFID ↓ | Code utilization | FID ↓ | Precision ↑ | Recall ↑ |
|---|---|---|---|---|---|
| $4 \times 4 + 8 \times 8$ | 18.65 | 7524 + 8192 | 37.59 | 0.49 | 0.38 |
| $4 \times 4 + 8 \times 8 + 16 \times 16$ | 2.48 | 1954 + 7763 + 8192 | 15.66 | 0.61 | 0.54 |
| $8 \times 8 + 16 \times 16$ | 3.14 | 8008 + 8192 | 13.59 | 0.67 | 0.52 |
| $8 \times 8 + 16 \times 16 + 32 \times 32$ | 0.65 | 2656 + 8192 + 8192 | 11.10 | 0.64 | 0.58 |

outperforms ViT-VQGAN ($32 \times 32$) although we admit that ViT-VQGAN can be implemented more efficiently based on the hierarchical code maps.

**Ablation study on HQ-VAE** Table 4 supports the rationale of our choice, showing that resizing with pixel-unshuffle and -shuffle outperforms the average pooling followed by the nearest interpolation by 0.53 in FID since it prevents information loss during the downsampling operation. We also observe that longer training boosts rFID performance from 3.14 to 2.61. Note that learnable resizing turns out to be not effective, where we adopt a simple network with single convolution and transposed convolution layers with kernel size 2 and stride 2, for downsampling and upsampling, respectively. We impose the same loss function on the reconstructed images using only the top codes to avoid top code collapse. Refer to the supplementary document for a more detailed analysis.

**Ablation study on HQ-Transformer** Table 5 compares the effect of decoding policies and highlights the benefit of the proposed locally hierarchical decoding. VQ-VAE-2 [8] first generates the entire top codes and then derives the bottom codes one by one with global conditioning on the sequence of the top codes. RQ-Transformer [17] sequentially generates a top code and the relevant bottom codes, locally conditioned on the generated codes from the previous steps. We observe that our decoding strategy achieves better FID than the sequential methods since error accumulation during sampling is alleviated by our parallel decoding of bottom codes. We also validate the effects of soft-labeling as an alternative to one-hot labeling in the cross-entropy loss function for training HQ-Transformer as proposed in RQ-Transformer [17]. The use of soft labels with temperature $\tau = 1$ in our case improves performance as similarly observed in [17].

**Multi-level extensions** Table 6 present that three-level architectures outperform the two-level counterparts. For these experiments, we train two different combinations of three-level codes $\{(4 \times 4), (8 \times 8), (16 \times 16)\}$ and $\{(8 \times 8), (16 \times 16), (32 \times 32)\}$. The results show that the use of three-level codes is helpful for improving performance at the early stage of training (15 epochs for training HQ-VAE). However, they fail to achieve additional gains with more iterations probably due to the lower usage of top codes as shown in the "code utilization" column of the table. Detailed architectures for multi-level extensions are described in the supplementary document.

## 5 Conclusion

We presented a two-stage auto-regressive model, which consists of HQ-VAE and HQ-Transformer, to generate images based on hierarchically quantized codes. HQ-VAE represents an image by two-level discrete codes with different spatial resolutions, where it encodes the high-level structure of an image using the top codes and the low-level information using the bottom codes. HQ-Transformer models the multi-level codes efficiently and the combination of the two modules successfully generates unseen images. Experiments confirm the capability and potential of our model for high-quality and high-resolution image generation with a fraction of computational cost compared to existing methods.

**Acknowledgement** This work was partly supported by IITP grant [No.2022-0-00959, (Part 2) Few-Shot Learning of Causal Inference in Vision and Language for Decision Making, 2021-0-01343, Artificial Intelligence Graduate School Program (Seoul National University)] and NRF grant [No.2022R1A5A708390811, Trustworthy Artificial Intelligence] funded by the Korean government (MSIT).

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
