# Locally Hierarchical Auto-Regressive Modeling for Image Generation
## *Supplementary Document*

## A    Implementation Details

### A.1    HQ-VAE

For designing HQ-VAE, we modify the encoder and decoder architectures of VQ-GAN [A1]; we set the stride of the first convolution layer in the encoder and the last transposed convolution layer in the decoder as two as mentioned in the main paper. The encoder takes an image $\mathbf{x}$ and returns encoded feature map $\mathbf{z}$. We adopt HQ-VAE equipped with pixel-unshuffle and -shuffle for resampling. We use HQ-VAE $(16 \times 16)$ with $f = 16$ for the two-level HQ-TVAE implementation, which gives concise visual codes for efficient training of HQ-Transformer.

We train HQ-VAE $(16 \times 16)$ for 50 epochs on the ImageNet training split. We use the Adam optimizer with $(\beta_1, \beta_2) = (0.9, 0.5)$ and batch size 128. We set the learning rate $4 \times 10^{-5}$ with a warm-up for 0.5 epochs starting from a zero learning rate.

### A.2    HQ-Transformer

Table A presents the hyperparameters for HQ-transformer, where $d_e$ and $n_e$ denote the embedding dimension and the number of heads of the transformer, respectively.

Table A: Hyperparamters for implementing HQ-Transformer.

| Dataset | Model | # params | $N_{\text{IET}}$ | $N_{\text{MT}}$ | $N_{\text{PHT}}$ | $L_{\text{top}}$ | $L_{\text{local}}$ | $d$ | $d_e$ | $n_e$ | $\tau$ |
|---|---|---|---|---|---|---|---|---|---|---|---|
| | HQ-Transformer (S) | 530 M | 1 | 12 | 4 | 64 | 5 | 256 | 1536 | 24 | 1.0 |
| ImageNet | HQ-Transformer (M) | 870 M | 1 | 24 | 4 | 64 | 5 | 256 | 1536 | 24 | 1.0 |
| | HQ-Transformer (L) | 1437 M | 1 | 42 | 6 | 64 | 5 | 256 | 1536 | 24 | 1.0 |
| CC-15M | HQ-Transformer (S) | 579 M | 1 | 12 | 4 | 128 | 5 | 256 | 1536 | 24 | 1.0 |

The input of the main transformer starts with the start-of-sentence (SOS) token. A class-conditional image generation employs a class-specific SOS token while a text-conditional image generation uses a sequence of text tokens given by the byte pair encoding [A2, A3]. The length of text tokens is at most 64.

We train HQ-Transformer for 100 epochs on the ImageNet training split and 20 epochs on CC-15M, the aggregated training splits of CC-12M and CC-3M. Training HQ-transformer (S), (M), and (L) on ImageNet takes 35, 45, and 75 minutes per epoch with four NVIDIA Tesla A100 GPUs, respectively, while training HQ-transformer (S) on CC-15M requires 452 minutes per epoch in the same GPU environment. We use HQ-VAE trained on ImageNet for experiments on class- and text-conditional image generation. We use AdamW with $(\beta_1, \beta_2) = (0.9, 0.95)$, batch size 512, and weight decay coefficient $1 \times 10^{-4}$. We adopt two policies for learning rate schedule, warm-up and cosine annealing. At the first epoch, learning rate is warmed up gradually from $\text{lr}_{\text{init}} = 1 \times 10^{-5}$ to $\text{lr}_{\text{peak}}$. In the remaining training epochs, we adopt the cosine annealing from the peak learning rate $\text{lr}_{\text{peak}}$ to zero. We set $\text{lr}_{\text{peak}}$ to $5 \times 10^{-4}$ for ImageNet and $3 \times 10^{-4}$ for CC-15M and clip gradients at 1.0 for stable training.

Figure A and B demonstrate the performances of the baseline and rejection sampling by varying hyperparameters such as top-$k$, softmax temperature, and acceptance ratio. For the baseline sampling in ImageNet, the hyperparameter setting with $k = 2048$ and temperature $t = 0.95$ achieves the best FID performance in the small and medium models and the second-best performance in the large model. When the rejection sampling is employed in ImageNet, the setting with $k = 8192$,

36th Conference on Neural Information Processing Systems (NeurIPS 2022).

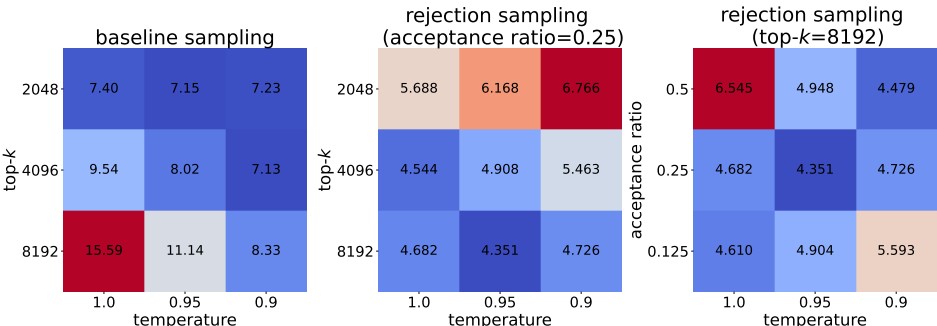

Figure A: The performance of HQ-TVAE (L) on the ImageNet training split in terms of FID with diverse configurations with top-$k$, softmax temperature, and acceptance ratio.

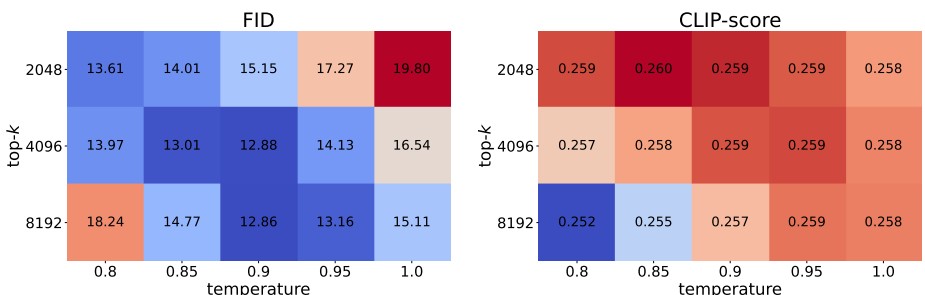

Figure B: The performance of HQ-TVAE (S) on the CC-3M validation split measured by FID and CLIP-score with diverse set of (top-$k$, softmax temperature).

temperature $t = 0.9$ and acceptance ratio $0.25$ achieves the best FID score. The setting with $k = 8192$ and temperature $t = 0.9$ for the baseline sampling leads to the best FID performance and comparable CLIP-score on the CC-3M validation split.

### A.3 Libraries and Code Repository

Our implementation is based on PyTorch 1.10 [1] with CUDA 11.3 and PyTorch Lightning 1.5.7 [2]. The major routines of our codes are based on the code repositories of VQ-GAN [3], minDALL-E [4], and RQ-Transformer [5].

## B Ablation study

### B.1 Details of Resizing Layers in HQ-VAE

We present the detailed analysis of down- and up-sampling operations on HQ-VAE and its effect on HQ-TVAE in Table B. When learning the resizing operations, we apply two different loss functions, pixel-wise $\ell_2$ reconstruction loss and perceptual compression loss to the reconstructed image based only on the top codes. According to our experiments, without such appropriate loss functions, the images only with top codes collapse as illustrated in the top of Figure C.

We observe that both reconstruction FIDs with multi-level codes and top code only rFID, denoted respectively by rFID and rFID-top, should be strong for achieving good image generation performance. Unfortunately, both metrics have the trade-off in most cases but the resizing based on pixel-shuffle is more effective for better image generation. Note that the convolution-based resizing operations

---

[1] PyTorch (https://pytorch.org)

[2] PyTorch Lightning (https://www.pytorchlightning.ai)

[3] VQ-GAN (https://github.com/CompVis/taming-transformers)

[4] minDALL-E (https://github.com/kakaobrain/minDALL-E)

[5] RQ-VAE and RQ-Transformer (https://github.com/kakaobrain/rq-vae-transformer)

Table B: Ablation study about down- and up-sampling operations in HQ-VAE and their impacts on the performances of HQ-TVAE.

| Resizing operation | Epochs | rFID ↓ | rFID (top) ↓ | FID ↓ | Precision ↑ | Recall ↑ |
|---|---|---|---|---|---|---|
| Conv2 + Deconv2 (Perceptual compression loss) | 15 | 3.72 | 18.36 | 15.68 | 0.67 | 0.52 |
| Conv2 + Deconv2 (Pixel-wise reconstruction loss) | 15 | 3.10 | 98.29 | 16.23 | 0.66 | 0.51 |
| Avg. Pooling + NN interpolation | 15 | 3.67 | 240.97 | 21.87 | 0.60 | 0.52 |
| PixelShuffle + Unshuffle | 15 | 3.14 | 83.45 | 13.59 | 0.67 | 0.52 |
| PixelShuffle + Unshuffle | 50 | 2.61 | 71.27 | 11.03 | 0.70 | 0.55 |

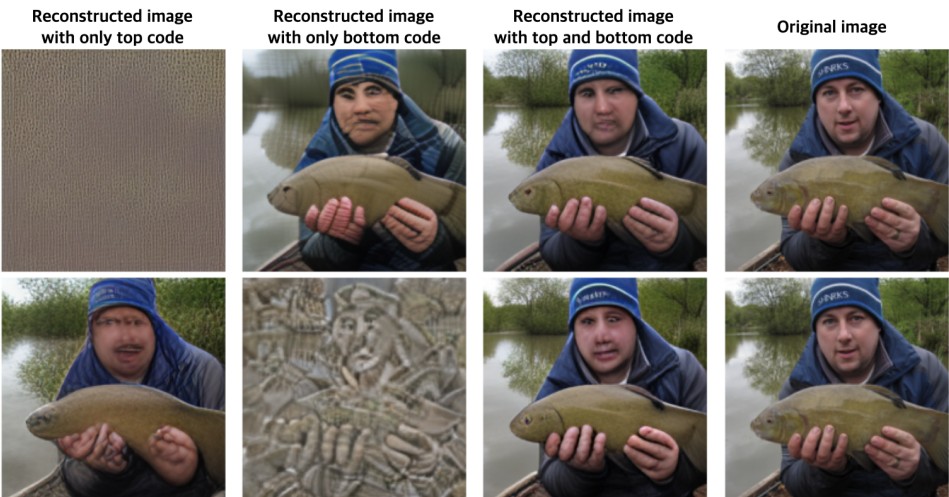

Figure C: Examples of reconstructed images using HQ-VAE with the learnable down- and up-sampling layers. Reconstructed images at the top are given by HQ-VAE trained without the reconstruction losses applied to top codes images while the images at the bottom are with the reconstruction losses. Note that the reconstruction only with top codes shown at the top-left corner fails in this example while the image at the bottom-left corner exhibits reasonable quality with high-level contents.

requires extra training time; HQ-VAE with the above learnable down- and up-sampling layer takes about 300 minutes per epoch, while HQ-VAE with pixel-shuffle takes about 170 minutes per epoch with four NVIDIA Tesla A100 GPUs.

## B.2 Input Embedding of HQ-Transformer

The ablation study in Table C(a) analyzes the effects on input embedding. We test two types of input embedding: addition and the input embedding transformer (IET) in (10) of the main paper. The addition method fuses the embeddings of top and bottom codes and the position embedding of the top code, which is given by

$$\mathbf{u}_i = \mathbf{e}_i^{\text{top,in}} + \text{Concatenation}\left(\mathbf{e}_{(i,1)}^{\text{bot,in}}, \cdots, \mathbf{e}_{(i,r^2)}^{\text{bot,in}}\right) + \text{PE}^{\text{top}}[i], \quad i > 0, \tag{1}$$

where $\text{Concatenation}(\cdot, \cdots, \cdot)$ denotes the channel-wise concatenation for bottom code embeddings. We set the number of self-attention blocks in IET to 1 or 2, $i.e.$, $N_{\text{IET}} = 1$ or 2. We observe that a single layer for IET outperforms the other settings in terms of FID and recall.

## B.3 Prediction Head Transformer (PHT)

We propose locally hierarchical decoding in PHT contrary to the standard sequential approach by assuming the conditional independence among bottom codes given a top code. Specifically, the locally hierarchical decoding first sample a single top code and then predicts all bottom codes at the same time conditioned on the top code. The ablation study Table C(b) demonstrates the benefit of our decoding strategy in the PHT with respect to image generation quality. Note that the proposed locally hierarchical decoding also has an advantage in speed.

Table C: Ablation study on the architecture of HQ-Transformer. We use the smallest model HQ-Transformer (S) to verify architectural choices.

| | Input embedding | Decoding policy | Label type | (top-$k$, $t$) | FID ↓ | Precision ↑ | Recall ↑ |
|---|---|---|---|---|---|---|---|
| (a) | Addition
IET $N_{\text{IET}} = 1$
IET $N_{\text{IET}} = 2$ | Locally hierarchical conditioning | One-hot label | $(2048, 0.9)$
$(2048, 0.95)$
$(2048, 1.0)$ | 11.03
10.01
11.86 | 0.70
0.69
0.70 | 0.55
0.55
0.51 |
| (b) | IET $N_{\text{IET}} = 1$ | Locally hierarchical conditioning
Locally sequential conditioning
Without local conditioning | One-hot label | $(2048, 0.95)$ | 10.01
11.33
26.72 | 0.69
0.68
0.52 | 0.55
0.55
0.59 |
| (c) | IET $N_{\text{IET}} = 1$ | Locally hierarchical conditioning | One-hot label
Soft-label ($\tau = 1$) | $(2048, 0.95)$ | 10.01
9.36 | 0.69
0.69 | 0.55
0.55 |

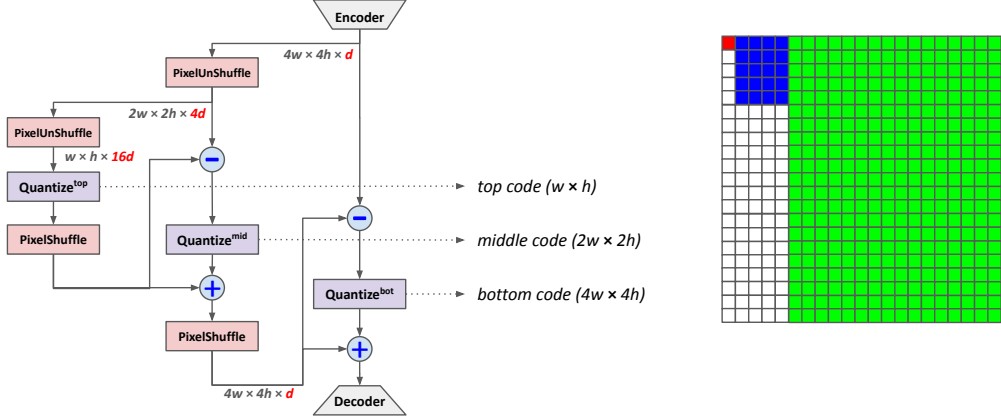

(a) HQ-VAE with three-level codes.

(b) Self-attention mask for three-level locally hierarchical decoding

Figure D: Multi-level extension for HQ-VAE (a) and self-attention mask for three-level locally hierarchical decoding in HQ-Transformer (b). We use pixel-shuffle and -unshuffle for resizing operations as illustrated in (a) while recursively quantizing hierarchical feature maps to acquire three-level codes—top, middle, and bottom. We also extend a self-attention mask to support locally hierarchical decoding for three-level codes as shown in (b).

### B.4 Soft-Labeling in HQ-Transformer

Table C(c) shows that soft-labeling improves FID compared to one-hot labeling.

### B.5 Multi-Level Extension of HQ-TVAE

We extend the default two-level HQ-VAE into a three-level network. Figure Da presents the architecture of three-level HQ-VAE while Figure E illustrates how the three-level HQ-VAE disentangles information into top, middle and bottom codes.

Locally hierarchical autoregressive model for three-level codes is easily extended from the two-level case. We introduce an additional conditional independence assumption among bottom codes given the associated top and middle codes for the multi-level extension, which is given by

$$
\begin{aligned}
&\mathbb{P}_\theta(t_i, \mathbf{m}_i, \mathbf{b}_i | \mathbf{t}_{<i}, \mathbf{m}_{<i}, \mathbf{b}_{<i}) \\
&= \mathbb{P}_\theta(\mathbf{b}_i | t_i, \mathbf{m}_i, \mathbf{t}_{<i}, \mathbf{m}_{<i}, \mathbf{b}_{<i}) \cdot \mathbb{P}_\theta(\mathbf{m}_i | t_i, \mathbf{t}_{<i}, \mathbf{m}_{<i}, \mathbf{b}_{<i}) \cdot \mathbb{P}_\theta(t_i | \mathbf{t}_{<i}, \mathbf{m}_{<i}, \mathbf{b}_{<i})
\end{aligned}
\tag{2}
$$

where $\mathbf{b}_i = (b_{(i,1)}, \cdots b_{(i,4r^2)})$ and $\mathbf{m}_i = (m_{(i,1)}, \cdots m_{(i,r^2)})$. This multi-level extension of (6) of the main paper is simply given by applying an additional chain rule to (5) of the main paper for introducing the middle code. HQ-Transformer with three-level codes is implemented by plugging in a self-attention mask for three-level locally hierarchical decoding in HQ-Transformer as illustrated in Figure Db.

We set the hyperparameters for decoding as follows: $k = 2048$ and temperature $(0.8^0, 0.8^1, 0.8^2)$ for three-level codes with resolution $(4 \times 4 + 8 \times 8 + 16 \times 16)$ and $(0.9^0, 0.9^1, 0.9^2)$ for three-level codes with resolution $(8 \times 8 + 16 \times 16 + 32 \times 32)$. The generation performance of the three-level

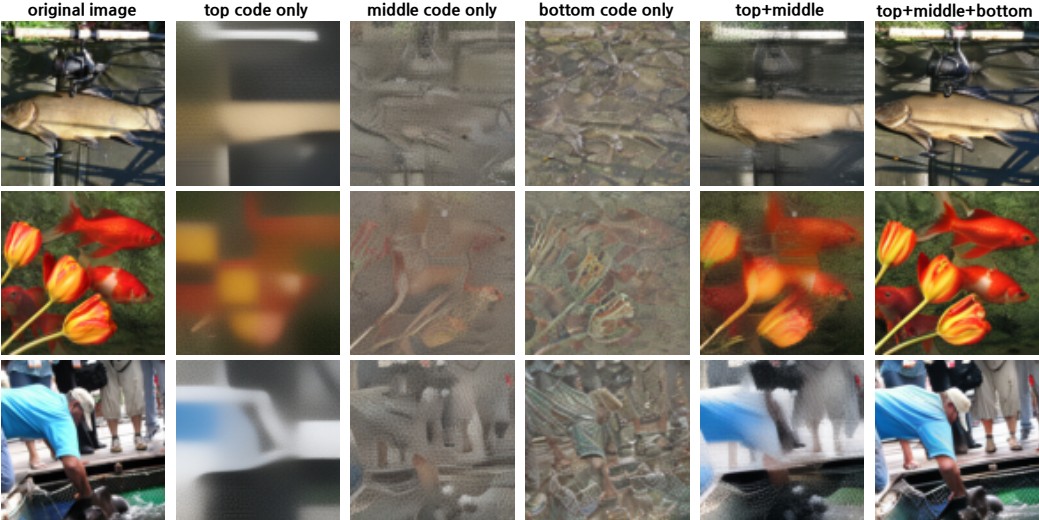

Figure E: Visualization of reconstructed images by HQ-VAE with three-level codes ($4 \times 4 + 8 \times 8 + 16 \times 16$). The first column shows the original image and the remaining columns show reconstructed images from top code only, middle code only, bottom code only, top+middle, and all the three-level codes. We can observe information disentangled in the top, middle and bottom codes in the visualization.

HQ-TVAE is improved by using lower temperatures on middle and bottom codes, but fails to reach the performance of the two-level counterpart.

## C   Additional Image Generation Examples

We demonstrate additional samples for class- and text-conditional image generation in Figure F and G, respectively. In both figures, a set of examples in three consecutive rows show the reconstructed images based on top, bottom, and two-level codes. All the generated samples in each class and each prompt are acquired by fixing a random seed to 0.

Based on generated samples from class- and text-conditional image generation, we also conduct top code-conditional image generation shown in Figure H and Figure I. Figure H shows top code-conditional image generation by both class- and text-conditional generated images. Figure I shows 23 generated images from only the single given top code, which demonstrates the diversity in detail generation of our models.

## D   Limitations and Negative Societal Impacts

### D.1   Limitations

Two-stage auto-regressive (AR) models have some artifacts in a small region with details. Figure J shows failure cases due to a scene with a small size of the face. Synthesized images only from the top code in Figure J contain the solid color of the face rather than the rough structure of a human face, such as mouth, nose, or eyes. Solid color on the face without the structure might degrade details created with the independence assumption. To solve this problem, more information should be injected into the top code without distortion even in the small region. Note that other two-stage AR models [A1, A4, A5] also experience the similar problems as reported in their supplementary materials. We believe that introducing expressive visual codes is still a promising solution to this problem.

The three-level extensions of our approaches have limitations over two-level baselines. First, the three-level HQ-VAE has lower effective top code usage, which results in poor performance compared to its two-level counterparts. Second, the three-level model requires tuning additional hyperparameters for locally hierarchical decoding.

## D.2   Negative Societal Impacts

Unknown biases or unfiltered inappropriate samples in large-scale datasets might contribute to the generation of biased or harmful content and our method also shares the core of these issues. We believe that more research is needed on those problems for large-scale image generation.

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

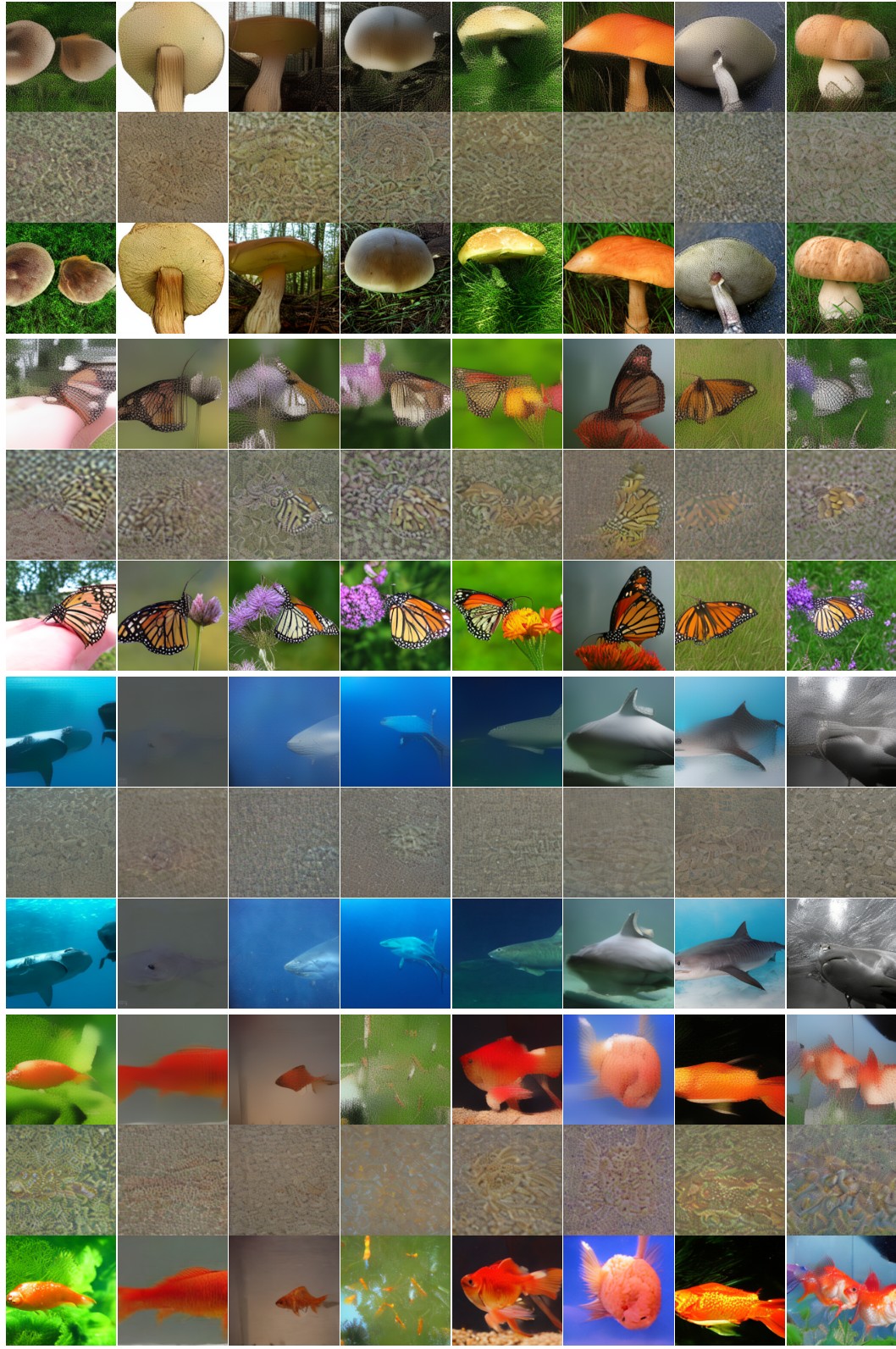

Figure F: Examples of class-conditional image generation with an ImageNet class *hen-of-the-woods* (996), *monarch butterfly* (323), *tiger shark* (3) and *goldfish* (1). The first row shows samples synthesized only from the top codes, the second row from the bottom codes only, and the third row shows our final generation samples.

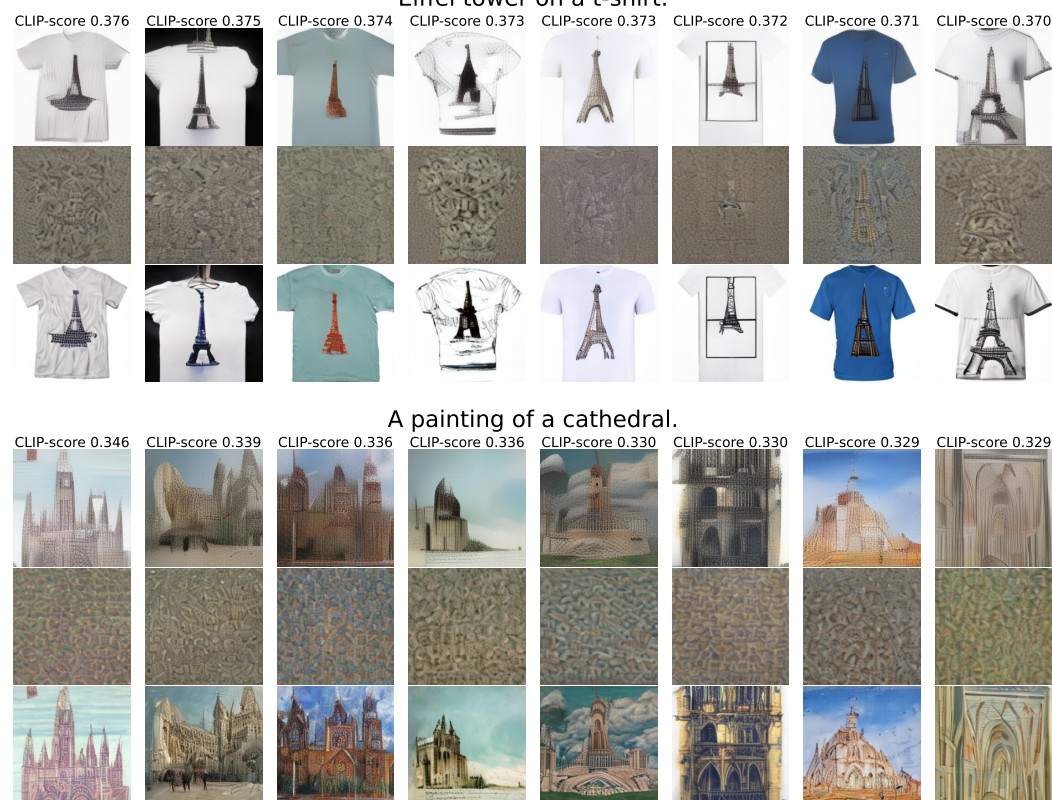

Figure G: Examples of text-conditional image generation from text prompts written in the top of each case. We visualize 8 samples out of 128 for each prompt using CLIP re-ranking. At the top of each column, we present the CLIP score between an image and a text prompt. The first and second rows show the images synthesized only with the top and bottom codes, respectively, and the third row illustrates the generated images with both the codes.

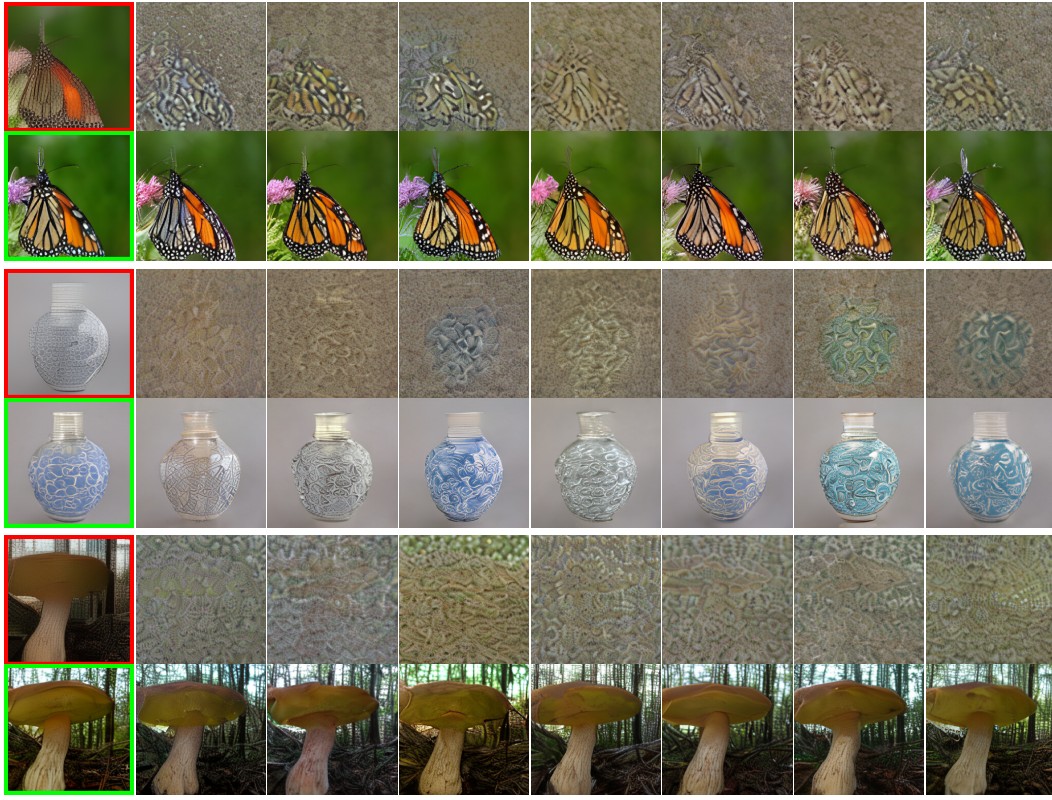

Figure H: Examples of top code-conditional image generation with generated images by conditioning ImageNet classes *monarch butterfly* (323), *vase* (883), *hen-of-the-woods* (996) shown in top-to-bottom. The images at the top-left corner in red bounding boxes visualize images synthesized from the top codes only while the images in green bounding boxes are corresponding images with full codes. The images except the first column are the generated images by varying the bottom codes with the top code fixed.

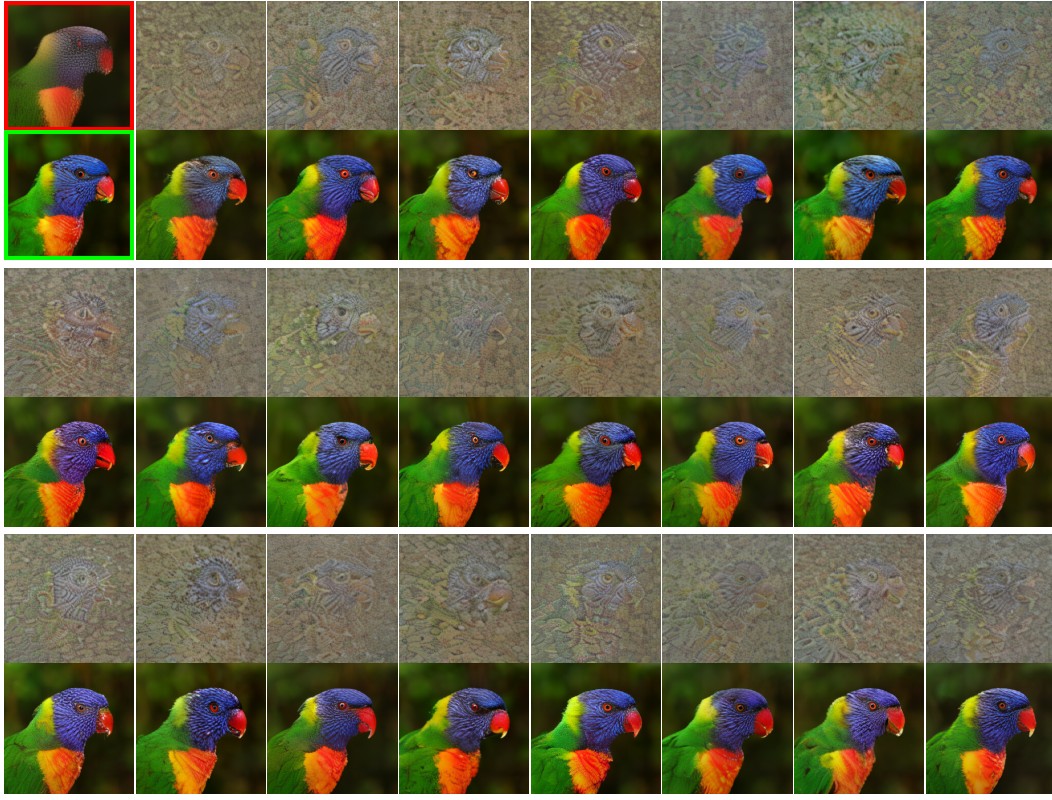

Figure I: An example of images generated using the identical top code in an ImageNet class *lorikeet* (90). The image in red bounding boxes visualize the image synthesized from the top codes only while the image in green bounding boxes are corresponding images with full codes. The remaining examples show the generated images by varying the bottom codes with the same top code. Note that every example exhibits various details under the same structure corresponding to the top code.

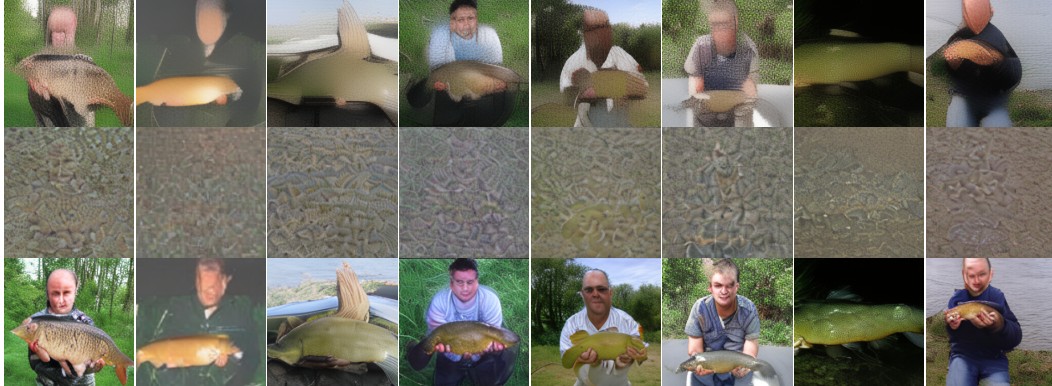

Figure J: A failure case of class-conditional image generation from an ImageNet class *doctor fish* or *tench* (0). The first and second rows show the images synthesized only with the top and bottom codes, respectively, and the third row illustrates the generated images with both the codes. The structure of human face are severely distorted in the first row images.