# OpenReview forum: "Locally Hierarchical Auto-Regressive Modeling for Image Generation"
_NeurIPS.cc/2022/Conference — NeurIPS 2022 Accept_

### Official Review · Reviewer_skYU · 2022-07-11

**Rating:** 6
**Confidence:** 5
**Soundness:** 3 good
**Presentation:** 2 fair
**Contribution:** 3 good

**Summary:**

In this paper, a hierarchical method for synthesizing natural images based on discrete representations is presented. More precisely, a combination of the previous two-stage hierarchical VQVAE-2 [1] and residual vector quantization (RQ-Transformer [2]) is first used to learn a hierarchical two-stage discrete latent space (called HQ-VAE). In a second phase, as usual, a generative prior is learned over the latent space. For this purpose, the "HQ-Transformer" is introduced, which, given a top code, predicts all corresponding local bottom codes in parallel. Experiments show good results in class-conditional modeling of ImageNet and in text-image synthesis of conceptual captions with improved sampling speed over related two-stage discrete methods.

**Questions:**

Please see the above section for questions.

**Limitations:**

Yes, limitations and societal implications are discussed, albeit very briefly.

**Strengths And Weaknesses:**

**Strengths**

- The proposed approach is simple (which is good!).
- In my opinion, the parallel local decoding is interesting and quite promising for further upscaling of the model.
Side question: for the parallel decoding scheme, is it possible to make use of a resampling strategy that uses the most confident codes only akin to [3]?
- The study on avg.-pooling vs channel-resampling is nice and the paragraph on using learnable down/upsampling layers here should be moved to the main paper.

**Weaknesses**

- The main drawback of the paper is it somewhat incremental nature over RQ-Transformer[2] and VQVAE-2 [1]. The main novelty seems to be the parallel decoding scheme,  while the increase in sample quality is limited and still contains lots of perceptual artifacts. I am curious about the other reviewers' opinion on this.
- While the comparisons to other "off-the-shelf" models in Tab.1-3 is interesting, I think the more interesting comparison would be to compare to a VQVAE-2 style hierarchical code (which can be trained with VQGAN reconstruction losses) and the strictly hierarchical autoregressive bottom -> top sampling scheme proposed there **in the same setting** (16x16 + 8x8, 8192+8182). In a related manner, what are the autoregressive decoding parameters used for the comparison in Table 5?
- For text-to-image: My understanding is that the model is directly conditioned on text embeddings obtained from BPE. Does the model further improve by deploying a fixed or learnable (potentially pretrained) text encoder here?

_Minor:_
- some of the samples in Fig.7 show lots of artifacts which are often associated with an overly reliance on perceptual losses such as LPIPS. Does weighing this term down in the reconstruction loss improve upon this? Related: Does the proposed HQ-VAE structure make training the discriminator more difficult?
- Is it possible to add more levels to the model (e.g. training a top-middle-bottom model)? Can this be used to train models with even higher receptive field in the bottom codes (e.g., for further improving sampling speed)?
- l. 196: And.we --> And we
- l. 199: casual self-attention --> causal self-attention

_Summary:_

This paper presents a simple and straightforward extension of RQ-Transformer and VQVAE-2 to learn a hierarchical two-stage discrete generative model of images. I like the general setup, the parallel (local) decoding strategy is good and a reasonable assumption, and although it is a straightforward combination of existing approaches, I think this approach has some potential to produce really good samples at a decent sampling speed. Further questions: How well does the model scale to even higher resolutions? Training the model on 32x32 and 16x16 latent codes would be interesting. In my opinion, the main drawback of the paper are both its limited increase in quality of the samples and somewhat limited novelty. This is why I currently view the paper as a borderline submission, but I am happy to be convinced otherwise by the authors and/or other reviewers.


_Literature_:
- [1]: Generating Diverse High-Fidelity Images with VQ-VAE-2, Razavi et al
- [2]: Autoregressive Image Generation using Residual Quantization, Lee et al
- [3]: MaskGIT: Masked Generative Image Transformer, Chang et al

---

> ### Author Response · Authors · 2022-08-02
> **Response to Reviewer skYU**
>
> **Parallel Decoding with Non-AR**
> Thank you for sharing an interesting idea. Considering the effectiveness of the resampling strategy shown in MaskGIT [3], we think that it can also improve our parallel decoding scheme.
>
>
> **Ablation study on resampling operations**
> We will move the paragraph on the ablation study of resampling operations into the main paper.
>
>
> **On the novelty of the proposed framework**
> We appreciate your constructive comments to clarify the novelty of our approach. First of all, we remark that the main contribution lies in two parts:
> - HQ-VAE leverages a locally hierarchical approach to effectively disentangle the information of different levels, as the top codes contain the global-level information and the bottoms encode only detailed local information. A proper design of resampling operations maximizes the information contained in top and bottom codes, while minimizes the information overlap between top and bottom codes. Thus, our model enables us to assume that the bottom codes could be conditionally independent given their associated top code.
> - Based on this assumption, HQ-Transformer can leverage the parallel decoding scheme, which is unavailable to previous studies, over the bottom codes, while generating an image in a locally hierarchical manner.
>
> **On the parallel decoding scheme**
> Our parallel decoding scheme assumes that several bottoms codes associated with the top code could be generated in parallel, rather than an autoregressive manner. As the reviewer’s comment, the parallel decoding is the key contribution of our study to make our model generate an image faster than previous studies. Here, we remark that the parallel decoding is feasible, since our hierarchical quantization module can alleviate the dependency on the bottom codes given by the top code. In addition, Table 5 shows that this parallel decoding is competitive with previous AR models in terms of the fidelity of generated samples.
>
> In the response to the artifacts in Figure 7, we suspect that this behavior appears due to the existence of watermarks. Please refer to the separate response below.
>
>
> **Comparison to VQ-VAE 2**
> We regret that it is infeasible for us to train VQ-VAE 2 combined with VQ-GAN losses due to the short rebuttal period. Nonetheless, we train VQ-VAE 2 style strictly hierarchical model (from top to bottom) on the same HQ-VAE and compare them with HQ-Transformer:
> | Model |   FID   | Precision | Recall | Throughput [samples/second] |
> | ------------------------------------ | ------ | ------- | ------------- | -------- |
> | VQ-VAE 2 (paper)                           | ~ 30   |               |            | - |
> | HQ-VAE + VQ-VAE 2 style (12+4)  | 23.25 |    0.61    |  0.53   | 45.00 |
> | HQ-VAE + VQ-VAE 2 style (12+8)  | 19.17 |   0.65    |  0.51    | 9.78   |
> | HQ-VAE + HQ-Transformer (12+4)      | 11.34 |    0.71    |  0.52   | 78.87 |
>
> Consequently, HQ-Transformer outperforms the strictly hierarchical models as shown in the table. Moreover, throughputs of strictly hierarchical models are much worse than HQ-Transformer, since the lengthy bottom code remains the main bottleneck. This shows that our decoding scheme based on a locally-hierarchical fashion is suited to HQ-VAE, better than VQ-VAE 2.
>
> We include the experiment details for completeness. AR model for top code has 12 layers, 24 heads, and 1,536 dimensions. For the bottom code sampling conditioned on the top, we train two AR models with 4 and 8 layers to show the effect of depth in transformer. The number of heads and dimensions for bottom code models are 24 and 1,536, respectively. The maximum throughput is reported by the measurement of several batch sizes {100, 200, 500, 1000, 2000} on a single A100.
>
>
> **Sampling parameters in Table 5**
> We used the top-k of 2048 and top-p of 1.0. The detailed version of ablation study in Table 5 is shown in Appendix (Table 2 and ll. 66 - 95).
>
>
> **Language models on T2I generation**
> We agree with the reviewer’s comment that a pre-trained text encoder can further improve the performance of text-to-image generation. As shown in a recent study [a], incorporating large-scale language models in HQ-Transformer is worth exploring for future work.

---

> > ### Author Response · Authors · 2022-08-02
> > **Response to Reviewer skYU (2)**
> >
> > **On the cause of artifacts in Fig. 7**
> > We think that the artifacts in Fig.7 appear due to the watermark images rather than the limitation of our approach since about 10.6% of images in CC3M and CC12M contain various types of watermarks. To estimate the statistics of watermark images, we use the publicly available watermark classifier [b] of the threshold 0.8.
> >
> >
> > **Multi-level extension**
> > We use hierarchical codes with two-level (4 x 4 + 8 x 8) and three-level (4 x 4 + 8 x 8 + 16 x 16) to validate multi-level extension within the rebuttal period. For detailed modifications to the architecture of HQ-VAE and HQ-Transformer, refer Figure 11 in the revised Appendix.
> > This is a reasonable setting because two settings exploit the equivalent sequence length (4 x 4) in the transformer body and show similar throughput each other.
> > The comparison of stage 1 (reconstuction FID) and stage 2 model (FID, Precision, Recall) is shown in the below. Stage 1 model is trained on ImageNet with 15 epochs:
> >
> > | Code Resolution          | rFID    | FID     | Precision | Recall | Throughput |
> > | ----------------------------- | --------- | --------- | ------------ | -------- | --------------- |
> > | 4 x 4 + 8 x 8                 | 18.65  | 37.59   |  0.49       |  0.38   |      93.46     |
> > | 4 x 4 + 8 x 8 + 16 x 16 | **2.48** | **18.95** |  **0.57**   | **0.54** |      89.29    |
> >
> > Adding more hierarchical layers improves the reconstruction FID (rFID) in stage 1 model and generation quality (FID, Precision, Recall) in stage 2 with similar throughput. We expect that the perceptual performance and throughput tendancies still holds in the case of (8 x 8 + 16 x 16 + 32 x 32).
> >
> >
> > - [a] Photorealistic Text-to-Image Diffusion Models with Deep Language Understanding, 23 May 2022, ArXiv, https://arxiv.org/abs/2205.11487
> > - [b] https://github.com/LAION-AI/watermark-detection

---

> > > ### Comment · Reviewer_skYU · 2022-08-07
> > > **Thanks**
> > >
> > > Thank you for your detailed answer and the additional experiments that were possible during the rebuttal period. Since I think that this approach, despite limited novelty, offers an interesting possibility to optimize discrete image models especially in terms of decoding speed, I give it the benefit of the doubt and increase my score to 6. However, to have a really significant impact I recommend, if possible, to train it on a larger scale, i.e. one downsampling less.

---

### Official Review · Reviewer_4ntj · 2022-07-11

**Rating:** 7
**Confidence:** 4
**Soundness:** 3 good
**Presentation:** 3 good
**Contribution:** 3 good

**Summary:**

The paper proposes a hierarchical autoregressive transformer model for image generation. Based on a quantizing autoencoder (VQGAN/VQ-VAE) the paper splits the quantized representation into two groups called top and bottom codes. The top codes have a smaller spatial resolution **r** (e.g., 8x8) and encode high-level, abstract information. The bottom codes have a larger spatial resolution **2r** (e.g., 16x16) and encode more detailed texture/high-frequency information. Each bottom code is considered to be only dependent on one top code, i.e., each top code is "responsible" for 4 bottom codes. During generation, the top codes are sampled autoregressively, however, the bottom codes can be sampled in parallel (conditioned on the respective top code) which significantly boosts the sampling speed. Results on class- and text-conditional image generation show similar or improved results compared to recent SOTA baselines while having a significantly higher sampling throughput than other autoregressive models.

**Questions:**

I would also be interested in the model's performance on unconditional generation or other conditioning, e.g., label map to image or similar.

Also, do you think the performance or resolution could be further improved by adding even more hierarchical layers of quantized codes?

**Limitations:**

One of the main limitations still remains as the top codes still have to be sampled autoregressively. Other approaches, e.g. MaskGIT, sample all codes in parallel and then iteratively resample code entries with low probability which overall also reduces the numbe of model evaluation steps. I wonder if approaches like that could be combined with the current approach?

Also, according to table 3 the size of the latent representation does affect the reconstruction capability so it is not clear how well this scales to larger resolutions.

**Strengths And Weaknesses:**

The paper tackles an important problem of autoregressive image generation models, i.e., their sampling speed. The proposed approach makes the assumption that the high-level image content can be modeled via a relatively small spatial resolution while detailed texture information can be modeled locally without relying on information from far away (in pixel space). This hierarchical approach allows faster sampling speeds due to parallel generation of the texture codes while the high-level image codes (which still have to be sampled autoregressively) can be represented in a smaller spatial dimension.

Overall the paper is well written and motivated. I believe the methodology section could be slightly improved to make it easier to understand. Maybe Fig 2 can be slightly extended to show more details of the transformer. I believe the encoder/decoder structure is quite clear and easier to understand from the textual description.

---

> ### Author Response · Authors · 2022-08-02
> **Response to reviewer 4ntj**
>
> **Experiments on an unconditional generation task**
> We will report the performance of HQ-Transformer in the unconditional generation task on FFHQ during the author-reviewer discussion period.
>
>
> **Multi-level extension**
> Adding more hierarchical layers improves the quality of image reconstruction in the stage 1 (HQ-VAE) and image generation in stage 2 (HQ-Transformer).
> To show complete results within the rebuttal periods, we fix top code resolution into 4 x 4 for faster validation. Specifically, we use configurations with two-level codes (4 x 4 + 8 x 8) and three-level codes (4 x 4 + 8 x 8 + 16 x 16). We remark that this setting can fairly compare the two configurations, since the two settings have a similar throughput, while using the same sequence length, (4 x 4), for the transformer body. We attach the implementation details of adding more hierarchical levels in Figure 11 in the revised Appendix.
> Thetable below shows the effectiveness of adding more hierarchical levels in terms of reconstruction FID (rFID) of HQ-VAE and FID, precision, and recall of HQ-Transformer. Here, HQ-VAE is trained on ImageNet for 15 epochs:
>
> | Code Resolution          | rFID    | FID     | Precision | Recall | Throughput |
> | ----------------------------- | --------- | --------- | ------------ | -------- | --------------- |
> | 4 x 4 + 8 x 8                 | 18.65  | 37.59   |  0.49       |  0.38   |      93.46     |
> | 4 x 4 + 8 x 8 + 16 x 16 | **2.48** | **18.95** |   **0.57**   | **0.54** |      89.29    |
>
> We expect the performance improvement above to be also applied in the case of (8x8 + 16x16 + 32x32).
>
>
> **Extension to non-AR modeling on top codes**
> Thank you for sharing an interesting idea for future work. We think that the current HQ-Transformer can be combined with the parallel decoding of MaskGIT, if our model is trained using the masked token modeling instead of autoregressive modeling. We think that combining MaskGIT with HQ-Transformer is a promising approach to further improve the sampling speed and the quality of image generation.
>
>
> **Higher resolution on the codes**
> Scaling up the resolution of latent codes improves the reconstruction quality because it prevents HQ-VAE from losing the information of input images. However, we remark that HQ-VAE with (16x16+32x32) codes produce enough reconstruction quality.
> Considering that Table 3 only contains the performance of stage 1 model,  the reviewer might be questioned on the generation performance of larger code map resolution. So, we include the additional experiment results on generation with larger code resolution settings (16 x 16 + 32 x 32). The table below shows that (16 x 16 + 32 x 32) setting improves precision and recall more than that of (8 x 8 + 16 x 16), where HQ-VAEs of both configurations are trained in 10 epochs for a fair comparison.
>
> | Code Resolution   | rFID |   FID   | Precision | Recall |
> | ------------------------ | ------ | ------- | ----------- | -------- |
> |  8 x 8 + 16 x 16     | 4.22 | 15.28 |    0.64      |  0.51   |
> |  16 x 16 + 32 x 32 | **0.85** | **13.63** |   **0.64**      |  **0.60**   |
>
> This is because the larger code resolution improves the reconstruction quality significantly, thereby achieving a better quality of generated samples.
> Note that our model provides a more scalable solution than the other AR models with comparable code resolution. For example, the computational complexity of HQ-Transformer with code resolution (16 x 16 + 32 x 32) model is much less than that of VQ-GAN with code resolution (32 x 32).

---

> > ### Author Response · Authors · 2022-08-09
> > **Response to reviewer 4ntj (2)**
> >
> > ### Unconditional image generation on FFHQ
> > Our model achieved comparable performance to RQ-Transformer in unconditional image generation on FFHQ with an image resolution of 256 x 256 pixels.
> >
> > | Model                 | Code Resolution   | rFID  | FID     |
> > | ---------------------- | ------------------------ | ------ | -------- |
> > | RQ-Transformer | 8 x 8 x 4                | 7.29  | 10.38 |
> > | Ours                   | 8 x 8 + 16 x 16      | 5.53  | 10.21 |
> >
> > In specific, we only substitute class-conditional start-of-sentence (SOS) tokens into a single SOS token in HQ-Transformer to conduct unconditional image generation FFHQ. We train HQ-VAE with 150 epochs and HQ-Transformer with 200 epochs. Codebook size is 8192 for each top and bottom code and HQ-Transformer has 1024 dimensions, 16 heads, 24 layers in transformer body, and 4 layers in prediction head transformer. Hyperparameters for optimization on FFHQ are equivalent to those for ImageNet.
> >
> > We expect further performance gain by finding better hyper-parameters and extensions to a multi-level setting. We leave it for future work due to the time limit.

---

### Official Review · Reviewer_CKx9 · 2022-07-11

**Rating:** 4
**Confidence:** 4
**Soundness:** 2 fair
**Presentation:** 3 good
**Contribution:** 2 fair

**Summary:**

This paper focused on the problem of improving computational efficiencies of auto-regressive (AR) models on high-resolution images. The authors proposed a hierarchical-quantized VAE which models the images at different scales and a Transformer model for the encoded sequence at each scale. Evaluations on class-conditional and text-conditional tasks between the proposed method and the baselines were shown.

**Questions:**

It would be helpful if the authors could address the issues and concerns listed above.

**Limitations:**

The authors provide some discussions on the limitation regarding the independence assumption. More discussions on how the proposed method would behave when the target resolution increases would be helpful.

The authors also provided discussion on potential negative social impact.

**Strengths And Weaknesses:**

## Strengths

1. The paper is generally nicely written and well-organized. In addition, the paper contains a good coverage of the related background and training details (architecture selection, hyper-parameter settings).
2. The paper included thorough ablation studies to validate the design decisions.

## Weaknesses

1. The proposed methodology seems incremental: representing images using VQ-VAE in a coarse-to-fine manner was previously explored (VQ-VAE2[28], HAM[10], Hierarchical VQ-VAE[22], RQ-VAE[18]). In particular, RQ-VAE[18] also disentangles the information among different code levels (depths) without specialized modules. In addition, [18] proposed RQ-Transformer which also models codes at different levels in an AR fashion. In terms of computation complexity, RQ-Transformer[18] has the same dominating factor as the proposed method as well.
2. There are some concerns regarding the generalizability of the proposed method:
    1. The empirical validation is insufficient for disentanglement of the latent code as only 2 scales were explored. It is also not clear how the proposed approach could be extended to more than 2 scales. Since the claimed contribution involves the disentanglement of information into different codes, having a general principle that could be extended to an arbitrary number of scales is important and more empirical evidence should be present.
    2. It is not clear whether the conditional independence assumption would still hold when $r>2$ which would be important as the target image resolution increases (e.g. generating images with 512x512 resolution and above).
3. The formulation in equation 6 rewrites the joint probability into the product of two terms. During inference, the proposed method maximizes the second term first (by drawing the best $t_i$) and then maximizes the first term based on the chosen $t_i$. This is a sub-optimal strategy as maximizing the second term and then maximizing the first term does not equal maximizing the joint probability. Since the paper presents this as an inference strategy, the corresponding drawbacks/tradeoffs of this approach should be analyzed and discussed.
4. There are some important details missing:
    1. How does the HQ-Transformer generate diverse samples when the top code is fixed? Does it sample $\bf{b}_0$ only or does it apply stochastic sampling for all $\bf{b}_n$ like RQ-Transformer[18]?
    2. There are some missing throughput results in table 1.
5. Some minor issues
    1. In equation 11: there is a duplicate $u_1$, it should be $u_0$ instead?
    2. What were the batch sizes used to time the throughput in Tables 1 & 2 for different models?

---

> ### Author Response · Authors · 2022-08-02
> **Response to Reviewer CKx9**
>
> **On the novelty of the proposed framework**
> Thanks for the constructive feedback. Similar to the mentioned studies, HQ-VAE also exploits the coarse-to-fine approximation of images to disentangle the information among different levels. However, compared with VQ-VAE 2 and RQ-VAE, we remark that our HQ-VAE is the indispensable part for achieving efficient image generation in Table 1, since we assume that the bottom codes are conditionally independent given the associated top code. We think that this assumption is not too strong since HQ-VAE enforces the top codes to contain the global-level information, while the bottom codes encode only detailed local information.
>
>
> **Multi-level extension of HQ-VAE and HQ-Transformer**
> A general principle can be applicable in extending HQ-VAE to an arbitrary number of scales. Specifically, we can construct multi-level HQ-VAE by recursively quantizing the hierarchical feature map between original feature map and aggregated quantized feature map from top code to current code level. For example, HQ-VAE with three-level of code resolution (4x4 + 8x8 + 16 x 16) is shown in Figure 11 in the revised Appendix.
> To validate multi-level extension within the time budget, we use configurations with two-level codes (4x4 + 8x8) and three-level codes (4x4 + 8x8 + 16x16). We think it is a fair setting due to two reasons. First, two configurations use the same code resolution (4x4) in the transformer body Second, they have comparable throughputs.
> The comparison of the performance of stage 1 (reconstuction FID) and stage 2 model (FID, Precision, Recall) is shown in below. We trained stage 1 model on ImageNet with 15 epochs equally:
>
> | Code Resolution          | rFID    | FID     | Precision | Recall | Throughput |
> | ----------------------------- | --------- | --------- | ------------ | -------- | --------------- |
> | 4 x 4 + 8 x 8                 | 18.65  | 37.59   |  0.49       |  0.38   |      93.46     |
> | 4 x 4 + 8 x 8 + 16 x 16 | **2.48** | **18.95** |   **0.57**   | **0.54** |      89.29    |
>
> Increasing the hierarchical levels of the codes improves the reconstruction quality in the stage 1 model. We also observe that disentanglement of HQ-VAE still works in multi-level HQ-VAE as shown in Figure 12 in the revised Appendix. Generation performances of three-level model, measured as FID, Precision and Recall, are better than those of bi-level model. We expect that the above results are also applied to the case with code resolution of (8x8 + 16x16 + 32x32).
>
>
> **On the conditional independence when r > 2**
> Although the conditional independence assumption empirically holds in the case of r=2, it may not be generalizable to r>2, since the global-level information encoded in the top code is not sufficient to explain all the local details spread over several bottom codes. Our suggestion for the high-resolution target is then to increase the number of hierarchical layers, while keeping r=2. For instance, HQ-VAE of the three levels configuration (4x4 + 8x8 + 16x16) still holds the conditional independence on the lower-level codes given the associated upper-level code, since this is the case of r=2. Please refer to the additional experiments on HQ-VAE of 4x4 + 8x8 + 16x16 configuration in the response above.
>
>
> **On the sub-optimal inference strategy**
> We appreciate your feedback on the inference strategy. In our inference procedure, the top codes are first generated by the standard multinomial sampling, rather than the greedy argmax sampling. Then, the bottom codes corresponding to the top are simultaneously decoded by the multinomial sampling. We respectfully disagree that this inference scheme is sub-optimal, since this still maximizes the joint distribution over the top and bottom codes, which is factorized in a locally-hierarchical fashion.
>
>
> **Top code-conditional sampling**
> HQ-Transformer can generate diverse samples with the fixed top codes, since the bottom codes are generated using the standard multinomial sampling, not using the greedy (argmax) sampling. Meanwhile, we do not use the stochastic sampling of RQ-Transformer during image generation, since it is a training strategy.
>
>
> **Missing throughputs in Table 1**
> We will update the throughput of MaskGIT and VQ-Diffusion in Table 1, since the official codes of these are publicly available.
>
>
> **Minor issues**
> Thank you for indicating the typo. It should be u0 and u1.
> We measured throughputs of RQ-Transformer (1.4B and 3.8B) and HQ-Transformer (S, M, L) on several batch sizes, and reported the best throughput. The batch sizes of HQ-Transformer (S, M, L) are 2,000, 1,000, and 500, respectively. The batch sizes of RQ-Transformer (1.4B and 3.8B) are 500 and 200. For the other models in Table 1, we borrowed the reported throughputs in LDM [a].
>
> [a] High-Resolution Image Synthesis with Latent Diffusion Models, CVPR 22

---

> > ### Comment · Reviewer_CKx9 · 2022-08-06
> > **Re: Response to Reviewer CKx9**
> >
> > ## Multi-level extension
> >
> > Thanks for the added information. In the three-level setting, what is the formulation of HQ-Transformer? Specifically, how would you modify Equation 6? Suppose now you have the top code, the middle code and the bottom code, do you assume conditional independency among the bottom codes given the associated middle code only, or is it also conditioned on the associated top code as well?
> >
> > ## Suboptimal inference strategy
> >
> > By “inference strategy” I meant “MAP inference” during training instead of the network forward pass during test-time. How do you maximize the first term in Equation 6 during training?
> >
> > ## Training Strategy Details
> >
> > For clarification, do you apply stochastic sampling in training HQ-Transformer? Are there any other training strategies you adopt for HQ-Transformer?

---

> > > ### Author Response · Authors · 2022-08-07
> > > **Response to Reviwer CKx9 (2)**
> > >
> > > ### Multi-level extension
> > >
> > > We use conditional independence assumption among bottom codes given the associated top and middle codes for the multi-level extension as shown in the equation below. This multi-level extension of Eq. 6 is simply given by applying the chain rule to Eq. 5 with the middle codes.
> > >
> > > **Multi-level extension of Eq. 6.**
> > >
> > > $$\\mathbb{P}_\{\\theta\} ( t_i, \\mathbf{m}_\{i\}, \\mathbf{b}_\{i\} | \\mathbf{t}_\{< i\}, \\mathbf{m}_\{<i\}, \\mathbf{b}_\{<i\} )
> > > = \\mathbb{P}_\{\theta\} ( \\mathbf{b}_i | t_i, \\mathbf{m}_i, \\mathbf{t}_\{< i\},  \\mathbf{m}_\{<i\}, \\mathbf{b}_\{<i\} )
> > > \\cdot \\mathbb{P}_\{\theta\} ( \\mathbf{m}_i | t_i, \\mathbf{t}_\{< i\},  \\mathbf{m}_\{<i\}, \\mathbf{b}_\{<i\} )
> > > \\cdot  \\mathbb{P}_\{\theta\} \( t_i | \\mathbf{t}_\{< i\}, \\mathbf{m}_\{<i\}, \\mathbf{b}_\{<i\} )
> > > $$
> > >
> > > where, $$ \\mathbf{b}_i = ( b_\{(i, 1)\} , \\cdots b_\{(i, 4r^2)\} ) \text{ and, } \\mathbf{m}_i = ( b_\{(i, 1)\} , \\cdots b_\{(i, r^2\)} )$$
> > >
> > >
> > > ### Suboptimal inference strategy
> > >
> > > In the first term of Eq. 6, we optimize the predicted probability of bottom codes conditioned on **the ground truth of associated top code** and **the ground truth of previous codes**. We can simultaneously optimize the first and second terms in Eq. 6, because we use the ground truth of codes.
> > >
> > > This strategy is so-called teacher forcing for AR models, which is a typical training strategy exploiting the ground truth of codes via maximum-likelihood estimation (MLE), instead of maximum a posteriori (MAP) estimation.
> > >
> > > Teacher forcing can be sub-optimal since the training process is exposed to the distribution of the sequence of codes with only ground truths, not an appropriate sampling distribution. Like typical AR models, we also adopt top-k, top-p sampling to mitigate the problem of teacher forcing. This is an inherited limitation of all auto-regressive models trained by teacher forcing, including our models.
> > >
> > >
> > > ### Training Strategy Details
> > >
> > > In training, we did not use stochastic sampling, but only used soft-labeling of temperature $\tau=1$ as an alternative to the one-hot labels in the cross-entropy loss function. We did an ablation study on the effect of soft-labeling, as shown in Table 5.

---

> > ### Author Response · Authors · 2022-08-09
> > **Response to Reviwer CKx9 (3)**
> >
> > ### Throughput of MaskGIT and VQ-Diffusion
> > We measure the throughput of MaskGIT [b] and VQ-Diffusion [c] based on their official code repositories and report the performance on the revised main paper. MaskGIT and VQ-Diffusion achieve the best throughputs with batch size 200. Two methods show lower throughput compared to our HQ-Transformer.
> >
> > Caching mechanism of the transformer is crucial for the throughput of auto-regressive (AR) models. With cached attention, the transformer only computes attention for a newly-sampled single code at each sampling step and skips exhaustive computation of attention for all codes. Contrary to AR models, Non-AR models, such as MaskGIT and VQ-Diffusion, exhaustively compute attention of all codes because these models update the entire codes at each step.
> >
> > - [b] MaskGIT: Masked Generative Image Transformer, CVPR 22, https://github.com/google-research/maskgit
> >
> > - [c] Vector Quantized Diffusion Model for Text-to-Image Synthesis, CVPR 22, https://github.com/microsoft/VQ-Diffusion

---

### Author Response · Authors · 2022-08-07
**Reminder for author-reviewer discussion period**

Welcome to discuss the proposed method or still unsolved questions after our answers. We still have more than a couple of days before the end of the author-reviewer discussion period.

Thank you for your constructive comments.

---

### Author Response · Authors · 2022-08-09
**Summary of the contributions of the proposed method**

### Summary of the contributions of the proposed method
We appreciate all reviewers for their efforts in constructive comments and discussions. We summarize the main contributions of the proposed method below.

- We propose a locally hierarchical auto-regressive (AR) model leveraging **a pyramid of discrete visual codes** for efficient image generation based on a two-stage AR framework.
- Stage 1 model HQ-VAE encodes an image into hierarchical codes by effectively disentangling the information of different levels, as the top codes contain the global-level information and the bottoms encode only detailed local information. A proper choice of resampling operations maximizes the contained information in the top and bottom codes, while **minimizing the overlapped information** between the top and bottom codes. Based on disentanglement in HQ-VAE, we introduce a conditional independence assumption on bottom codes given their associated top code.
- Stage 2 model HQ-Transformer leverages the conditional independence assumption into a **parallel decoding scheme**. The parallel decoding scheme generates locally hierarchical codes from top to bottom code level order and samples codes in each level in parallel. Propelled by the parallel decoding, our model samples fast with comparable performance to the other generative models.
- We validate the proposed framework can be extended into a **multi-level setting**. We demonstrate three-level setting with code resolution (4 x 4 + 8 x 8 + 16 x 16) and compare it with two-level setting with code resolution (4 x 4 + 8 x 8). We observe that the three-level setting outperforms the two-level setting.

---

### Meta-Review · Area_Chair_rAGF · 2022-08-25

**Recommendation:** Accept
**Confidence:** Less certain

**Metareview:**

The work proposes hierarchical generation as an approach to mitigate the increasing costs of modelling each pixel autoregressively as images increase to higher resolutions. The paper is easy to read, has strong empirical results, and has impressive ablation studies to understand the idea.

I think the paper could most be improved by clarifying its relationship to prior work, as well as being more specific about generalizability.

1. There are already quite a few works proposing hierarchical generation based on vector quantization such as VQVAE-2, RQ-Transformer, and RQ-VAE. The authors should include any discussion of differences in the paper. In the rebuttal, the authors argue the major difference is parallel decoding based on conditionally independent generation of bottom features given top. These approaches are conceptually so similar that they're worth comparing to in the experiments in a controlled setup, and not simply comparing paper numbers which use completely different setups like code size and number of parameters.

2. I share Reviewer CKx9's concern about the generalizability of the method as it's difficult to see how it could extend to higher resolutions than what it currently studies (256x256. The computational complexity analyzed in Sec 3.2.3 is incorrect as, say, the Prediction Head Transformer can dominate the complexity significantly more than the main Transformer that is assumed to be the most expensive in the analysis.

**Award:**

No

---

### Decision · Program_Chairs · 2022-09-14

Accept